# Metronidazole and ether derivatives target *Helicobacter pylori* via simultaneous stress induction and inhibition

Michaela K. Fiedler [1,10], Marianne S. I. Pandler[1,10], Ruolan Gong[2,3,4], Sonja Fuchs [2,3], Katharina Rox [5,6], Verena Friedrich[2,3], Dietmar Pfeiffer [2,3], Dharmesh Singh [2,3], Till Reinhardt[1], Cora Mibus [2,3], Matthias Huber[7,8], Corinna R. Hess[7,8], Raquel Mejías-Luque[2,3], Markus Gerhard [2,3], Michael Groll [9] & Stephan A. Sieber [1] ✉

Metronidazole is a front-line drug for the treatment of *Helicobacter pylori* infections. However, its mode of action and cellular targets are poorly defined, and higher dosing and combination therapies are required to overcome resistance. Here we performed activity-based protein profiling with tailored metronidazole probes and identified chaperonin HpGroEL and thiol peroxidase HpTpx as prominent targets, the latter being essential for *H. pylori* survival under oxidative stress. Alkynylated ether probes exhibited enhanced antibacterial potency compared with the parent drug in vitro, including activity against resistant strains. Biological assays, chemical proteomics and co-crystallization studies confirmed target engagement, with enhanced binding of ether derivatives to HpTpx. Refined ether analogues exhibited favourable pharmacological profiles without cytotoxicity. The in vivo activity of ether analogues using an *H. pylori* mouse model demonstrated full bacterial eradication at low dosing of 0.3 mg kg$^{-1}$ day$^{-1}$. Our findings reveal that stress induction and simultaneous inhibition of the stress response represent a mechanism of this compound class.

*Helicobacter pylori* is a Gram-negative, microaerophilic bacterium that colonizes the human stomach, causing numerous gastrointestinal disorders including gastritis, peptic ulcers and cancer[1,2]. A total of 43% of the world population is infected, and although often asymptomatic, it increases the probability of gastric cancer development and mucosa-associated lymphoma tissue owing to permanent inflammation[3]. A 7-day combination triple therapy consisting of two antibiotics that are metronidazole and clarithromycin and a proton pump inhibitor such as omeprazole used to be the front-line therapy to eradicate *H. pylori* in humans[4,5]. However, antibiotic resistance has decreased the efficacy of existing strategies against *H. pylori* infections within the past years[1,6]. A particular limitation of metronidazole is its moderate

activity against susceptible *H. pylori* strains with a minimal inhibitory concentration (MIC) range of 10–50 μM (refs. 2,7). Therefore, high dosing of the drug in combination with other adjuvants is needed to effectively eradicate *H. pylori*[5,8]. Although adverse side effects are considered tolerable, the likelihood increases in high-dose regimens and often involve the gastrointestinal tract by altering the composition of the microbiome[8–11].

A rational approach to re-establish antibiotic potency requires a comprehensive understanding of the cellular mode of action (Fig. 1a). Metronidazole, a 5-nitroimidazole prodrug, induces oxidative stress within the cells[12]. After (1) cell entrance, the nitro group is reduced, leading to (2) the formation of reactive radicals interacting with (3)

**Fig. 1 | Synthesis and biological evaluation of ABPP probes and 5-nitroimidazole derivatives. a**, The general mode of action of **Metro** in bacteria: (1) cell entrance via passive diffusion, (2) bioactivation of nitro group by respective electron-transport system and redox enzymes, (3) reaction of reactive radical species with cellular components such as DNA or proteins and (4) reoxidation to parent compound via 'futile cycling' in the presence of oxygen in aerobic bacteria. **b**, An overview of synthesized ABPP probes **Metro-P1**, **Metro-P2**, **Metro-P3** and **Metro-P4** and respective MIC values in *H. pylori* 26695. **c**, An overview of synthesized 5-nitroimidazole ether derivatives. **d**, The correlation plot of log MIC values to reduction potential determined via simple linear regression. **e**, A table showing MIC values of derivatives in *H. pylori* 26695 (*n* = 3) and their reduction potential $E_{red}$ against NHE. Dimetridazole-2-hydroxy and metronidazol are included for reference. Schematic in panel **a** partially created with BioRender.com.

different cellular components[13] (Fig. 1a). The activation process is complex and continues to be elucidated[9,14–17]. In *H. pylori*, the reductive activation probably requires oxygen-insensitive NADPH nitroreductases, including RdxA (mutations are most commonly associated with metronidazole resistance) and FrxA[18–20]. However, if oxygen is present, the toxic radical species are reoxidized to their (4) prodrug form, diminishing the bactericidal effects[12,21–23]. Thus, the presence of oxygen lends this approach suitable selectivity for antibiotic applications without distinct human cytotoxicity[24].

Despite the characterization of reactive intermediates, little is known about their exact targets in the cell. Metronidazole radicals generate oxidative stress and can react with proteins or DNA to cause manifold types of damage[12,13] (Fig. 1a). These interactions include covalent adduct formation with proteins, the inhibition of DNA synthesis and repair or DNA damage by oxidation[12,13,25–28]. *H. pylori* counteracts oxidative damage by tailored stress response enzymes such as HpGroEL, an essential chaperonin, which is required for the correct folding of proteins to repair damaged or denatured proteins[29–31]. Importantly, the thiol peroxidase (HpTpx), which is part of the thioredoxin (Trx) stress defence system is crucial for *H. pylori* to survive gastric oxidative stress by reducing toxic peroxides to water or alcohols, respectively[32,33]. Furthermore, previous studies revealed that HpTpx protects DNA from damage by ROS[34].

We describe here the synthesis, target identification, mechanistic analysis and in vivo efficacy of 5-nitroimidazole ether derivatives bearing 60-fold enhanced potency compared with metronidazole. Our study conceptualizes a dual mode of action by which metronidazole

and 5-nitroimidazole ethers induce stress and simultaneously inhibit the stress response with deleterious effects on bacterial survival.

## Results

### Design and synthesis of metronidazole probes reveal a boost of antibiotic activity independent of the redox potential

Metronidazole's core structure is based on an imidazole ring equipped with a nitro group at the 5-position, which is essential for its antimicrobial activity[12]. Four metronidazole activity-based protein profiling (ABPP) probes equipped with an alkyne tag were designed and synthesized. To minimally perturb the structural composition and preserve the nitro group, the handles were introduced at the 1- and 2-position as either ether or alkyl derivatives with or without an intact hydroxyl group (Fig. 1b and Supplementary Fig. 1).

Before labelling studies, all four probes were tested for their antibiotic efficacy against *H. pylori* 26695 in comparison with **Metro** as a reference (Fig. 1b). **Metro-P4**, a 4-nitroimidazole with an alkyl side chain, showed a decrease in antibiotic activity (MIC of 50 μM) compared with **Metro** (12.5 μM)[2,7] and was therefore not used in further ABPP studies. By contrast, whereas **Metro-P2**, resembling the free hydroxy group of **Metro**, showed a slightly lower MIC of 3.13 μM compared with the parent drug, **Metro-P1** and **Metro-P3** exhibited an unexpected 30- and 60-fold boost in activity with MIC values of 390 nM and 195 nM, respectively (Fig. 1b). These results highlight the ether moiety as an unprecedented alteration for enhancing antibiotic potency via an unknown mechanism. Moreover, **Metro-P3** exhibited superior activity compared with **Metro** against several metronidazole-resistant

*H. pylori* strains with substantially lower MICs in most of the isolates. Serial passaging showed no pronounced resistance development of 5-nitroimidazole ethers. Frequency of resistance (FoR) studies with **Metro-P3** at 12× MIC confirmed this notion (FoR < $2.01 \times 10^{-11}$); however, notable resistance occurred at 8× MIC (1.52 µM) (Supplementary Table 1 and Supplementary Fig. 2). As reductive activation without reoxidation to the parent compound of 5-nitroimidazoles can only occur under anaerobic or microaerophilic conditions, no biological activity was observed in aerobically grown *Escherichia coli* and *Staphylococcus aureus* for all compounds. Vice versa, anaerobic *Clostridium difficile* was susceptible to **Metro-P2** with a comparable MIC to **Metro** of 1 µM and decreased biological activity (MIC: 6.25 µM) for **Metro-P1** and **Metro-P3** (Supplementary Table 2).

On the basis of the discovery of the boost in activity, we synthesized several ether derivatives of **Metro**, **Metro-P1** and **Metro-P3** and tested them against two different *H. pylori* strains 26695 and PMSS1 (Fig. 1c). Interestingly, all ether derivatives showed the increase in activity, independent of their structural composition, such as the position or length of the ether chain (Fig. 1d and Supplementary Table 3). By contrast, 5-nitroimidazoles with aliphatic side chains exhibited a remarkably decreased biological activity (Supplementary Fig. 3). Yet, cyclic voltammetry measurements revealed no considerable correlation between redox potentials and MIC values of the ether derivatives (Fig. 1d,e; see Supplementary Table 3 for MIC values in micrograms per millilitre and Supplementary Fig. 4 for cyclic voltammetry plots), excluding enhanced reductive activation as the sole reason for the activity increase.

### Target identification studies reveal chaperonin HpGroEL and thiol peroxidase HpTpx as major hits

To identify the cellular targets responsible for the mode of action of **Metro** as well as the activity-boosted ether derivatives, we performed gel-based ABPP labelling studies in *H. pylori* live cells with **Metro-P1**, **Metro-P2** and **Metro-P3**[35,36] (Fig. 2a). Here, bacteria were grown to early stationary phase (Supplementary Fig. 5) before being treated with various probe concentrations, lysed and attached to rhodamine azide via click chemistry[37,38]. Labelled protein bands were visualized by fluorescent SDS–polyacrylamide gel electrophoresis (PAGE). Of note, distinct protein bands were observed, including a characteristic 18 kDa protein that was solely labelled by the ether probes, emphasizing a substantial difference in the target profile (Supplementary Fig. 6).

To unravel the identity of these target proteins, we performed mass spectrometry (MS)-based ABPP studies. In situ-labelled *H. pylori* proteome (1 µM of **Metro-P1**, **Metro-P2** or **Metro-P3**) was clicked to biotin azide, target proteins were enriched on avidin beads and tryptically digested peptides were analysed by liquid chromatography–tandem mass spectrometry (LC–MS/MS) via label-free quantification[39,40] (Fig. 2a). Identified proteins were visualized in volcano plots (Fig. 2b), and top hits were selected on the basis of an enrichment ratio of $\log_2 > 1.0$ and *P* value of <0.01 (*q*-value <0.05; Supplementary Table 5). Among those was an 18 kDa thiol peroxidase hit, which was subsequently confirmed via deletion of the corresponding *tpx* gene as the ether-specific signature gel-band (Supplementary Fig. 7).

To narrow down high-confidence targets, we performed competitive labelling experiments by pretreatment with varying excess of **Metro** upon addition of probe. The fluorescent SDS gel-based analysis showed a 60-kDa band decreased for all three probes, and again, the 18-kDa HpTpx band was solely diminished in case of ether derivatives **Metro-P1** and **Metro-P3** upon **Metro** pre-incubation (Fig. 2c). Interestingly, an in-depth analysis via LC–MS/MS confirmed thiol peroxidase HpTpx (18 kDa, O25151) and revealed another crucial stress-response protein, chaperonin HpGroEL (60 kDa, P42383), as significantly outcompeted targets matching the gel-based results (Fig. 2d and Supplementary Fig. 8). We extended our studies to other facultative anaerobic bacteria, which, similarly to *H. pylori*, rely on the Trx system

as the predominant oxidative stress response. Using **Metro-P3**, the labelling of GroEL and Tpx was confirmed in *Staphylococcus schleiferi* and *Staphylococcus pseudintermedius* under anaerobic conditions by both in-gel fluorescence and significant enrichment in MS-based proteomic analyses (Supplementary Fig. 9).

Gel-based competitive labelling assays were performed in *H. pylori* under the in situ microaerophilic environment to unravel apparent half-maximal effective concentration ($EC_{50}$) values for these hit proteins. The competition of **Metro-P3** binding with various excesses of **Metro** resulted in a high absolute $EC_{50}$ value of 250 µM for HpTpx (Fig. 2e and Supplementary Fig. 10). To compare this value with a more potent analogue of our collection, we selected **MF-03** (MIC of 390 nM; Supplementary Table 3) as a representative example with the ether substitution at N-1. Most importantly, **MF-03** showed strong competition of **Metro-P3** HpTpx binding with an $EC_{50}$ value of 17 µM (15-fold enhanced activity compared with **Metro**; Fig. 2f). By contrast, both compounds addressed HpGroEL, the second main target, with rather similar $EC_{50}$ values of 24 µM (**Metro**) versus 40 µM (**MF-03**) (Supplementary Fig. 11). These results highlight HpGroEL and HpTpx as major targets of both **Metro** and ether analogues; however, the strongly enhanced competitive binding of **MF-03** to HpTpx may account for the observed boost in activity.

### HpGroEL and HpTpx are covalently bound and inhibited by Metro and Metro-P3 after reductive activation

As binding of **Metro** and derivatives to HpGroEL and HpTpx requires the in situ activation of the nitro group under microaerophilic or anaerobic conditions, direct inhibition and binding assays with purified enzymes in vitro were challenging (Supplementary Figs. 12–14).

To alternatively validate binding and inhibition of the two main targets, we used a protein expression methodology that enables the in situ activation in living cells[26]. Thus, **Metro** or **Metro-P3** was directly added to the growth medium during the expression of recombinant HpGroEL or HpTpx in *E. coli* under anaerobic conditions to enable intracellular compound activation. During protein expression, time- and concentration-dependent modification of HpTpx in *E. coli* was observed (Supplementary Fig. 15) and binding of **Metro-P3** to HpTpx was confirmed via fluorescent SDS–PAGE (Supplementary Fig. 16). Intact protein MS (IP-MS) of labelled and purified HpGroEL and HpTpx revealed mass shifts of 141 Da (**Metro**) and 164 Da (**Metro-P3**) for both enzymes (Fig. 3a). Interestingly, a previously postulated mechanism[26,27] for the covalent reaction of 5-nitromidazole compounds with cysteines matches the observed loss of 31 Da due to reduction of the nitro to an amine group (Fig. 3b).

In the next step, we quantified the extent of the covalent modification of purified enzymes upon addition of varying concentrations of **Metro** and **Metro-P3** to the expression medium by IP-MS. In case of HpGroEL, we obtained almost equal binding of **Metro** and **Metro-P3** (0.9 fold; Fig. 3c). As HpGroEL has ATPase activity that is necessary for catalysing the protein folding of damaged or unfolded proteins[41], a malachite green-based ATPase assay was used to determine the influence of **Metro** and **Metro-P3** on the enzymatic activity. Indeed, both compounds impaired HpGroEL activity by 50% and 70% inhibition for **Metro-P3** (modified 71% of HpGroEL) and **Metro** (fully bound to HpGroEL), respectively (Fig. 3d,e). These data validate HpGroEL as a functionally inhibited protein target, which is comparably addressed by the amino forms of **Metro** and its probe **Metro-P3**.

When we performed the same concentration-dependent protein expression experiments with HpTpx in *E. coli*, **Metro-P3** resulted in a 3.6-fold higher HpTpx modification compared with **Metro** (Fig. 3f). To investigate the effect of reductively activated compound binding to HpTpx on its enzymatic activity, we carried out tailored peroxidase (Fig. 3g,h) and DNA damage rescuing assays (Supplementary Fig. 17). Both assays showed modification-dependent enzyme inhibition. For **Metro-P3**, we observed a strong inhibition of peroxidase activity

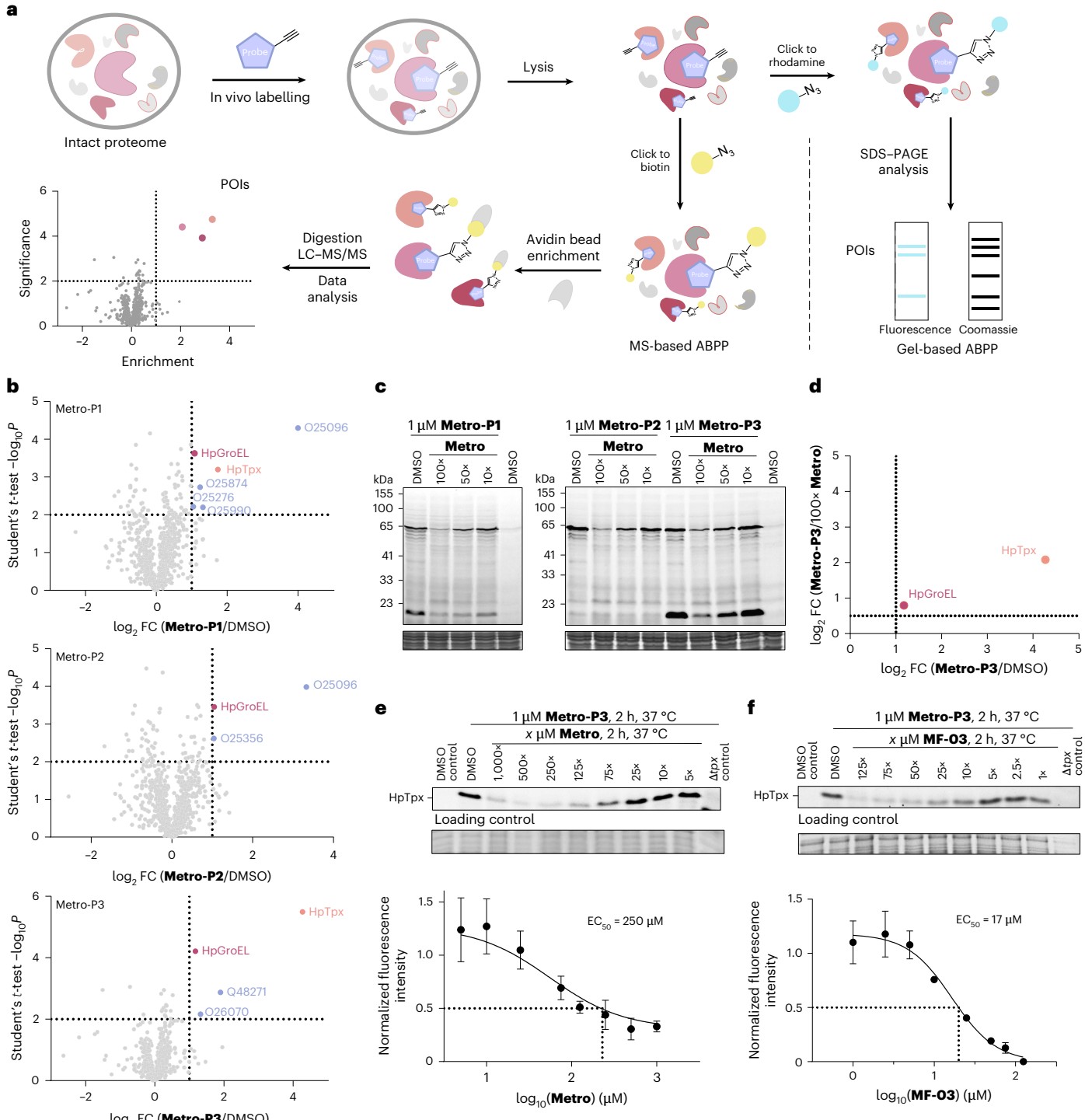

**Fig. 2 | Target identification studies of Metro in *H. pylori*. a**, The in situ gel-based or MS-based ABPP workflow using a **Metro** probe for the identification of proteins of interest (POI)[35,36]. **b**, The volcano plots with cut-off lines at $\log_2$ fold change (FC) of 1.0 and $-\log_{10} P$ value of 2 ($P < 0.01$) after labelling with 1 μM of each of **Metro-P1** (top), **Metro-P2** (centre) or **Metro-P3** (bottom) in *H. pylori* 26695 against DMSO. Enriched protein targets were marked in blue (Uniprot ID), in red (60 kDa chaperonin HpGroEL, Uniprot ID P42383) or in orange (thiol peroxidase HpTpx, Uniprot ID O25151). **c**, The gel-based competitive labelling in *H. pylori* with **Metro** before the addition of **Metro-P1**, **Metro-P2** or **Metro-P3** (1 μM). Results were reproduced in an independent repeat experiment. **d**, The outcompeted protein targets of **Metro-P3** displayed as a scatter plot with $\log_2$ FC of enriched proteins with **Metro-P3** (1 μM) or as competition with **Metro**

(100 μM). Proteins with $P$ values <0.01 and q values <0.05 in both conditions are depicted. **e,f**, Gel-based in situ competitive labelling in *H. pylori* 26695 for $EC_{50}$ determination: the pretreatment with **Metro** (1–1,000× excess) (**e**) or **MF-O3** (1–125× excess) (**f**) before incubation with **Metro-P3** (1 μM). The band intensities were quantified with ImageJ[70]. Values were normalized to band intensities without competitor, and DMSO control was subtracted as baseline. In **b** and **d**, experiments were performed in biological triplicates ($n_{bio} = 3$). A two-tailed Student's $t$-test was performed for statistics. In **e** and **f**, data represent mean values ± s.d. of three biological replicates ($n_{bio} = 3$). $EC_{50}$ values were determined using nonlinear regression and the extrapolation of $x$ values at 50% reduction (dotted lines). Panels **c**, **e** and **f** show fluorescence gel (top) and Coomassie gel (bottom) as loading control.

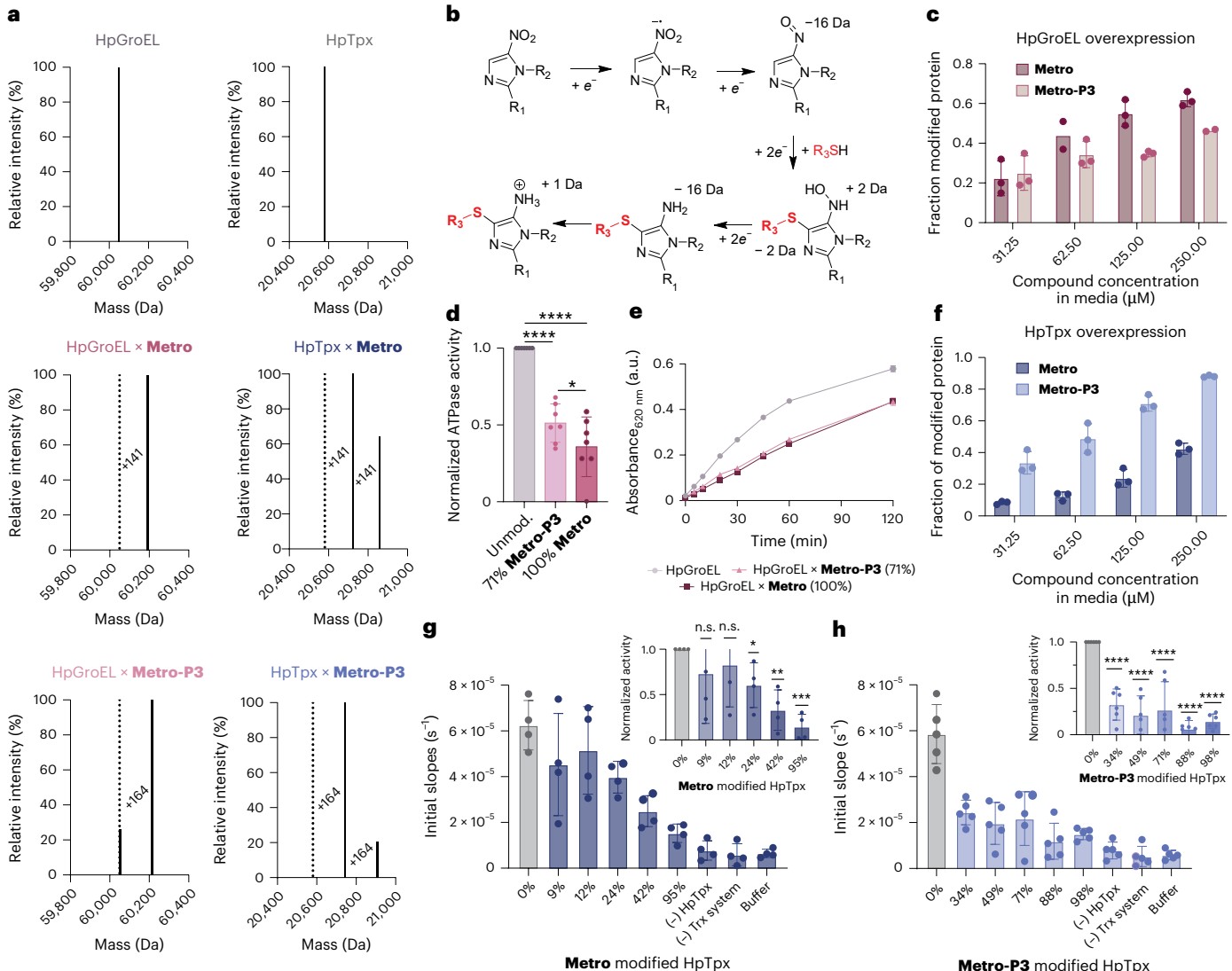

**Fig. 3 | Target validation studies of HpTpx and HpGroEL. a**, Mass shifts identified by IP-MS of HpGroEL and HpTpx modified with **Metro** (+141 Da) or **Metro-P3** (+164 Da) (loss of 31 Da of **Metro** and **Metro-P3** upon reductive activation). **b**, The proposed covalent adduct formation of 5-nitroimidazoles with cysteines[26,27] (Fig. 4a). **c**, Degrees of modification of HpGroEL with **Metro** or **Metro-P3** determined by IP-MS. Graph shows mean values ± s.d. of three measured IP-MS spectra per condition (n = 3). For 62.5 μM **Metro** and 250 μM **Metro-P3** only two data points could be measured. **d**, The ATPase activity was determined from the initial slope via linear regression of modified (71% **Metro-P3** or 100% **Metro**) HpGroEL and normalized to unmodified HpGroEL activity. Graph represents mean ± s.d. of 7 experiments (n_bio = 7). **e**, The malachite green assay evaluating HpGroEL activity. A control without HpGroEL was subtracted as baseline. Depicted are mean values of technical triplicates (n_tech = 3). **f**, The DoM of HpTpx with **Metro** or **Metro-P3** determined by IP-MS. Bar charts represent mean values ± s.d. of three measured IP-MS spectra (n = 3). **g,h**, The peroxidase activity assay of HpTpx modified with **Metro** (g) or **Metro-P3** (h). Rates of peroxide reduction coupled to the Trx system (Trx/Trx reductase) were determined by indirectly monitoring NADPH oxidation. Graph shows initial slopes determined by linear regression. Baseline was subtracted (−HpTpx) before normalization to unmodified HpTpx. Bar charts represent mean values ± s.d. of four (**Metro**) or five (**Metro-P3**) replicates (n_bio = 4–5). Statistical significance was determined on normalized (**d**) or unnormalized (**g,h**) data with one-way ANOVA with multiple comparisons (no correction). P values: >0.05 (n.s.), *P < 0.05, **P < 0.01, ***P < 0.001, ****P < 0.0001.

(70% reduced activity) already at 34% HpTpx modification. Of note, an eightfold higher concentration of **Metro** during protein expression was required to achieve a similar degree of modification with a comparable reduction in enzyme activity, highlighting a less pronounced effect of the parent drug on HpTpx.

## A previously unreported binding mode in HpTpx is exploited by 5-nitroimidazole ethers

To better understand the binding mode of both compounds, we performed binding site identification studies of **Metro** and **Metro-P3** for which we applied a chemical proteomics workflow in *H. pylori* using isotopically labelled desthiobiotin azide (isoDTB) tags[42] (Fig. 4a). Our analysis revealed a characteristic mass shift of **Metro-P3** bound via its

hydroxylamine derivative to C63 of HpGroEL and the active site cysteine C60 of HpTpx (Fig. 4b). Active site point mutations in recombinant HpTpx showed no binding to the C60A mutant and C60A–C94A double mutant, validating the results of the isoDTB workflow experiment (Fig. 4c).

To elucidate the molecular origin of the 5-nitroimidazole ether activity switch, we solved high-resolution X-ray structures of monomeric HpTpx and in complex with both ligands (see Supplementary Section 1 for detailed structural information). First, the structure of the ligand-free, reduced state (HpTpx^red) was determined at 1.75 Å resolution (PDB ID 9F5V), being composed of a five-stranded β-sheet flanked by four α-helices. A structural homology search using the DALI server[43] identified Tpx from *Yersinia pseudotuberculosis* (YpTpx) as the best hit

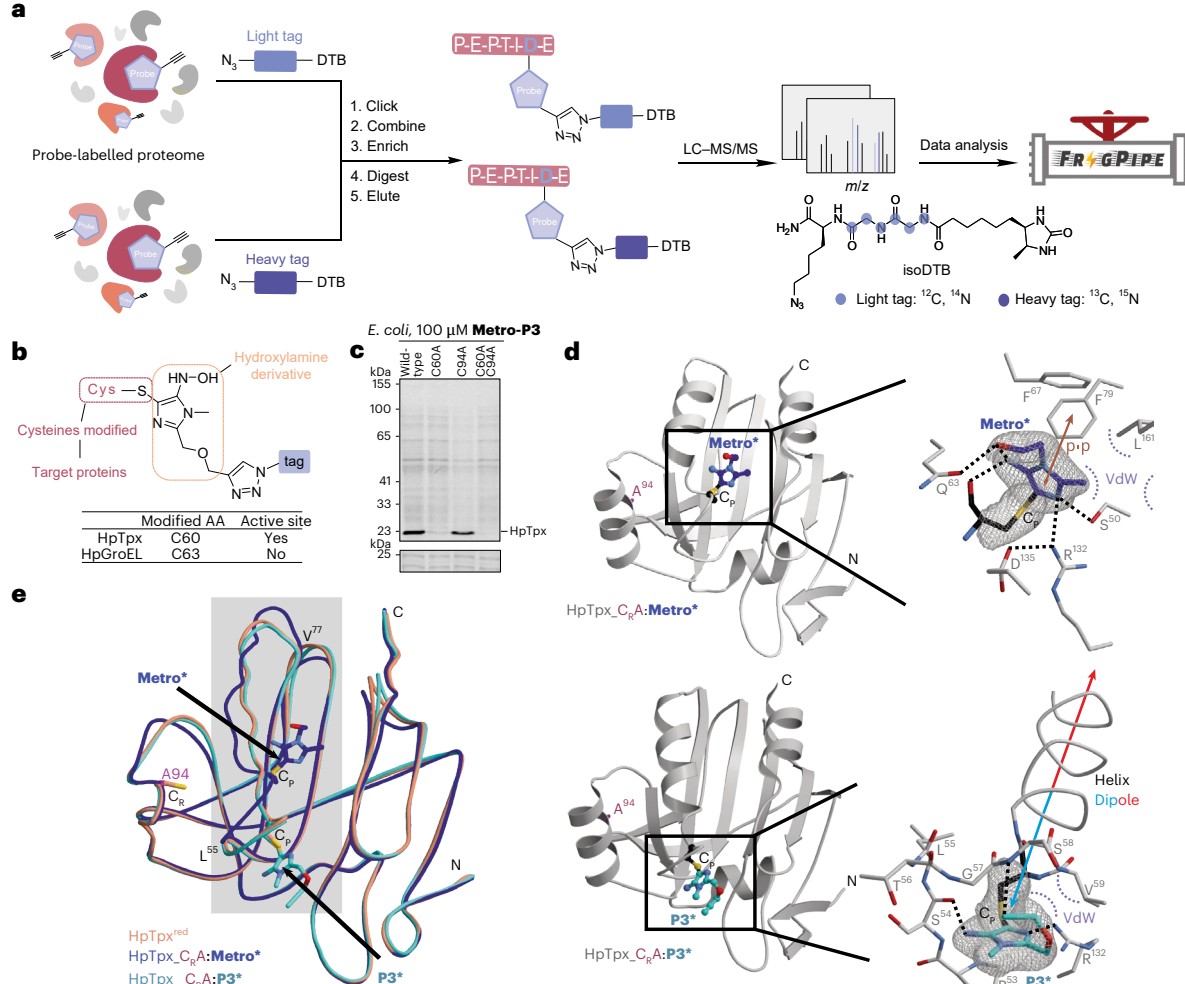

**Fig. 4 | Binding site identification of Metro and Metro-P3. a**, The binding site identification workflow using an isotopically labelled desthiobiotin azide tags proteomic workflow[42]. **b**, The binding site identification results of **Metro-P3**-labelled *H. pylori* proteome. AA, amino acid residue; Cys, cysteine; tag, heavy/light isoDTB tag. **c**, The gel-based labelling of recombinant HpTpx wild-type or mutant (C60, C94A, C60A-C94A) HpTpx in *E. coli*. Top: fluorescence gel. Bottom: Coomassie gel as loading control. Data are shown from one experimental replicate. **d**, The co-crystal structure of HpTpx C94A mutant (HpTpx_$C_R$A) with **Metro*** at 1.75 Å resolution (top) (PDB ID 9F64) and with **Metro-P3*** at 1.95 Å resolution (bottom) (PDB ID 9F65). Top: a ribbon diagram of the HpTpx_$C_R$A

mutant with **Metro*** (C atoms in purple). $C_P$ and $C_R$A are depicted in black and pink. The inhibitor is shown in a balls-and-sticks representation. Right: a close-up view of the active site (H bonds indicated by black dots; VdW, van der Waals). The $2F_o$–$F_c$ electron density map (grey mesh, contoured to confidence interval 1.0 $\sigma$) reveals full occupancy for **Metro***. Bottom: a ribbon structure of HpTpx_$C_R$A bound to **Metro-P3*** (**P3***, carbon atoms in cyan). The colour coding and orientation are according to the top. **e**, The superposition of HpTpx[red] (orange, PDB ID 9F5V) with HpTpx_$C_R$A:**Metro*** (purple) and HpTpx_$C_R$A:**Metro-P3*** (cyan). The covalent binding of **Metro*** results in notable structural rearrangements in the $C_P$ region. *Reduced amine form of nitro-prodrugs. DTB, desthiobiotin azide.

for comparison (PDB ID: 2XPD, sequence identity 39%)[44]. In HpTpx,[red] the peroxide-binding C60 ($C_P$) is located at the N-terminal part of helix α1 (Supplementary Fig. 18a). During catalysis, $C_P$ reacts covalently with hydroperoxides to form a cysteine sulfenic acid adduct ($C_P$-SOH). The subsequent condensation of $C_P$-SOH occurs via the resolving C94 residue ($C_R$) in helix α2, resulting in HpTpx[ox] with an intramolecular disulfide bridge between $C_P$ and $C_R$ (Supplementary Fig. 18b).

In the next step, we performed co-crystallization with HpTpx in the presence of **Metro** as well as **Metro-P3**. As the wild-type HpTpx did not co-crystallize, we incubated the HpTpx C94A mutant (HpTpx_$C_R$A) with **Metro** and **Metro-P3** during protein expression, which resulted in co-crystal structures of 1.75 Å (PDB ID: 9F64) and 1.95 Å resolution (PDB ID: 9F65), respectively (Fig. 4d,e). The crystal structure of HpTpx_$C_R$A in complex with **Metro** displays the inhibitor reduced to its amine derivative (**Metro***), which is fully defined in the observed electron density ($F_o$)–calculated electron density ($F_c$) map and forms an irreversible thioether bond with the $C_P$ residue (Fig. 4d, top). Although the superposition with HpTpx[red] exhibited identical conformations in the

mutated C94A region (Supplementary Figs. 18c and 19), notable structural rearrangements occurred in the ligand-bound structure at the catalytic centre of the $C_P$ residue. Helix α1 is shortened by two turns and distorted, and $C_P$ is shifted by 8 Å compared with HpTpx[red]. Intriguingly, **Metro** generated a well-defined specificity pocket at the active site that is absent in HpTx[red] as well as in YpTpx[ox] (Supplementary Fig. 18d). The interaction between the protein and **Metro*** was mediated by rearrangements of the aromatic amino acids Phe67, Phe79 and Tyr158 that stabilize the ligand's imidazole scaffold by π–π stacking (Fig. 4d, top).

Whereas **Metro-P3** formed the identical covalent thioether bond with $C_P$ upon reduction to its amine form (**Metro-P3***) (Fig. 4d, bottom), the HpTpx_$C_R$A–**Metro-P3*** complex surprisingly adopts a conformation as observed in HpTpx[red] (Supplementary Fig. 18e). Subtle shifts enable hydrogen bonds between the side chain of Arg132 and the main chain atom Ile152O as well as the N3 nitrogen of **Metro-P3**. The methylene moiety within the propargyloxy side chain of **Metro-P3** interacts via van der Waals contacts with the aliphatic residues Val59 and Leu153. Notably, the ether side chain is effectively coordinated by the helix

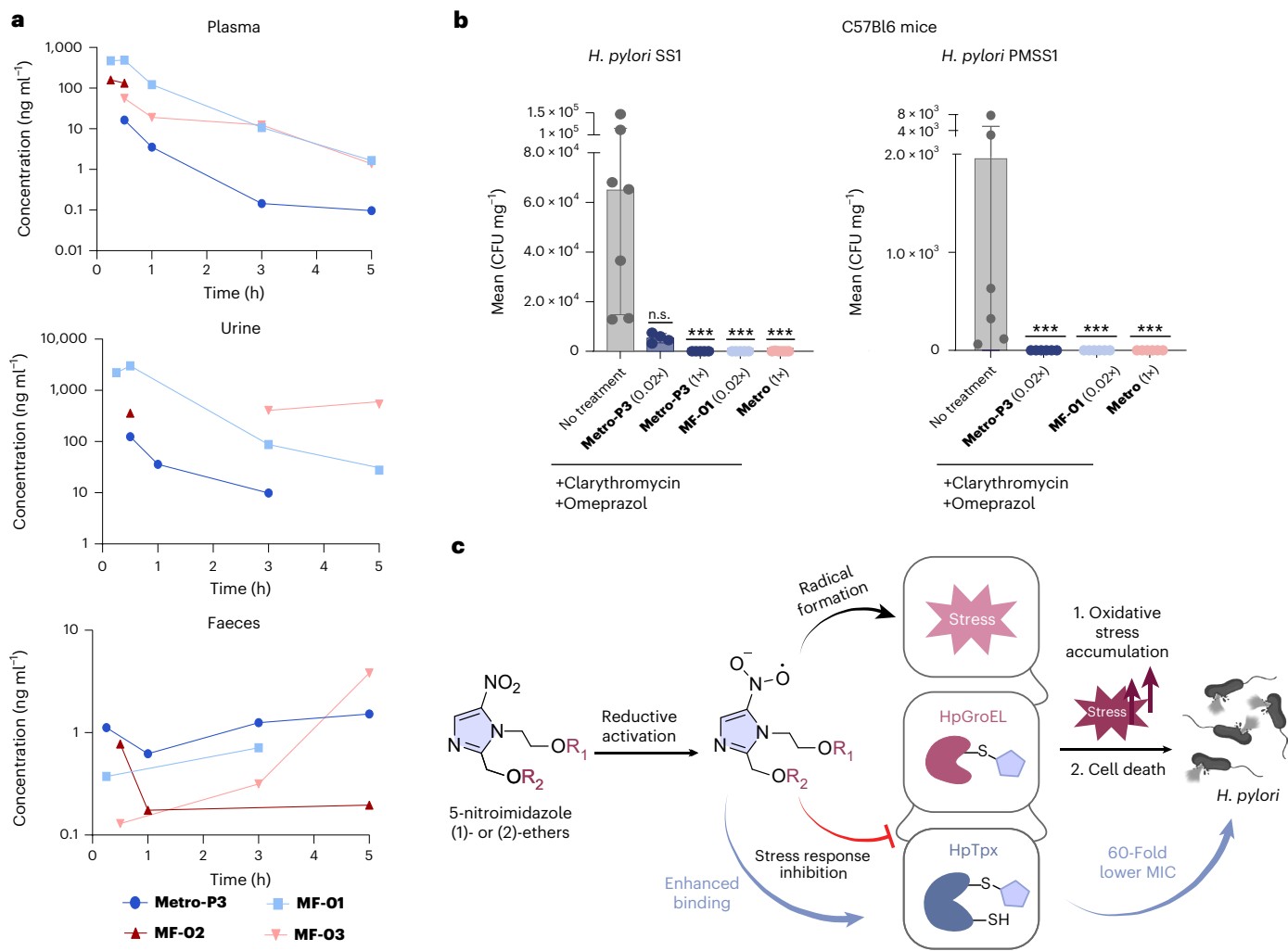

**Fig. 5 | Pharmacological profiling of 5-nitroimidazole ethers. a**, The PK data of **Metro-P3**, **MF-01**, **MF-02** and **MF-03** in plasma, urine and faeces (dose of 10 mg kg$^{-1}$ PO, $n$ = 2 mice per compound). **b**, In vivo efficacy studies in C57BL/6 mice infected with *H. pylori* SS1 strain and treatment with different regimens: no treatment ($n$ = 7), triple therapies consisting of clarithromycin (7.15 mg kg$^{-1}$ day$^{-1}$) and omeprazole (400 µmol kg$^{-1}$ day$^{-1}$) with low dose of **Metro-P3** (0.02× **Metro** = 0.30 mg kg$^{-1}$ day$^{-1}$; $n$ = 4) or **MF-01** (0.02× **Metro** = 0.30 mg kg$^{-1}$ day$^{-1}$; $n$ = 5), standard therapy with **Metro** (14.2 mg kg$^{-1}$ day$^{-1}$, $n$ = 6) or **Metro-P3** (1× **Metro** = 14.2 mg kg$^{-1}$ day$^{-1}$, $n$ = 5). The bar charts represent mean values ± s.d. Statistical significance was determined by non-parametric Kruskal–Wallis test

with multiple comparisons (no correction) to control column (no treatment). $P$ values: >0.05 (n.s.), ***$P$ < 0.001 (***). The LODs for this assay were calculated as 16.7 CFU mg$^{-1}$ (SS1) and 15.2 CFU mg$^{-1}$ (PMSS1), corresponding to one colony on the lowest dilution plated. **c**, The dual mode of action of 5-nitroimidazole ethers in *H. pylori*: upon reductive activation, reactive radicals increase stress within the cell, whereas the oxidative stress response is inhibited, eventually leading to *H. pylori* cell death. The enhanced binding affinity of 5-nitroimidazole ether derivatives to HpTpx led to a 50-fold activity boost in *H. pylori* in vivo. Schematic in panel **c** partially created with BioRender.com.

dipole moment of α1, at a proximity of 3.2 Å. This defined interaction aligns the entire inhibitor molecule within the active site and accounts for the enhanced binding affinity.

**MF-01 shows in vivo efficacy against *H. pylori* at low dosing**
On the basis of their promising potency, we evaluated the pharmacological profile of the 5-nitroimidazole ether derivatives in vitro. Cytotoxicity in human cells was determined by 3-(4,5-dimethylthiazol-2-yl)-2,5-diphenyl tetrazolium bromide (MTT) assays using two different cell lines (HeLa and HepG2), which revealed half-maximum inhibitory concentrations (IC$_{50}$) values above 1 mM for all compounds, enabling a sufficient efficacy window for further treatment of bacterial infections in vivo (Supplementary Table 3 and Supplementary Fig. 20). In line with these data, gel-based ABPP studies with **Metro-P1**, **Metro-P2** and **Metro-P3** in HeLa cells under aerobic conditions revealed no distinct labelling (Supplementary Fig. 21). In total, 9 out of 11 compounds exhibited excellent in vitro plasma stability with minimal degradation after 28 h of incubation (Supplementary Table 3). In addition, we could

show that antimicrobial activity translated into a co-culture model with infected adenocarcinoma cells where **MF-01** and **Metro-P3** expectedly showed a more noteworthy decrease in bacterial survival compared with **Metro** (Supplementary Fig. 22).

With these promising properties, six 5-nitroimidazole compounds (**Metro-P1**, **Metro-P3**, **MF-01**, **MF-02**, **MF-03** and **MF-07**) were subjected to in vitro absorption, distribution, metabolism, excretion (ADME) experiments. All molecules exhibited weak plasma protein binding (<75%), and four out of six compounds showed suitable ADME threshold values (Supplementary Table 4). In vivo PK studies in mice revealed favourable concentrations of **MF-01** to treat infections in the gastrointestinal tract with ~1.5-fold higher concentrations of **MF-01** in plasma and urine compared with **Metro-P3** (Fig. 5a; see Supplementary Fig. 23 for focused PK studies).

The efficacy of **MF-01** was evaluated in an *H. pylori* mouse infection model and compared with **Metro-P3** and **Metro**. Mice were infected with *H. pylori* SS1 for 4–6 weeks followed by the triple therapy (with clarithromycin and omeprazole) for 7 days. At 2 weeks after treatment, mice

were euthanized, and stomach homogenates were plated on Wilkins–Chalgren (WC)-Dent plates to determine CFU values. As expected, on the basis of the limited PK profile, **Metro-P3** eradicated *H. pylori* in vivo with a standard dosage comparable to **Metro** of 7.15 mg kg$^{-1}$ day$^{-1}$. In the case of treatment with **MF-01**, already 0.30 mg kg$^{-1}$ day$^{-1}$ was sufficient for full *H. pylori* eradication, a dosage at which **Metro** exhibited insufficient efficacy (Supplementary Fig. 24). This low dosing represents a major advance in the treatment of *H. pylori* infections in vivo, impressively reducing drug exposure (Fig. 5b). An analysis of the faecal microbiome showed that although standard triple therapy substantially reduced amplicon sequence variants (ASV) richness, single treatments with **Metro-P3** and **MF-01** largely preserved diversity at levels comparable to non-infected controls (Supplementary Fig. 25b). Moreover, we demonstrated that a suite of commensal murine gut bacteria has higher MICs values for our novel compounds **Metro-P3** and **MF-01** compared with **Metro** (Supplementary Table 6).

## Discussion

Despite the urgent need for more effective therapies against *H. pylori*, standard treatments mostly rely on the 60-year-old drug **Metro**, which has limited efficacy and requires high dosing in combination with other adjuvants. Deciphering a dual mode of action of this drug represents an opportunity for the development of improved antibiotics. Thus, the 60-fold boost in antibiotic activity in *H. pylori* of the here-discovered 5-nitroimidazole ethers is a remarkable advance towards lower dosing. This enhanced in vitro potency against *H. pylori* was a promising result that required further investigation of the molecular origin responsible for this triggered activity. Despite the known in situ formation of reactive radical species upon reductive activation of 5-nitroimidazoles, only two major proteins of the oxidative stress response, HpGroEL and HpTpx, were identified (Fig. 5c). Still, as proteomics is limited to protein identification, potential interactions with DNA or metabolites remain outside the scope of this analysis and therefore cannot be excluded. As direct validation of target proteins with recombinant enzymes in vitro was challenging, we favoured the native activation of 5-nitroimidazoles as a key step to evaluate their target preferences and deliberately avoided artificial activation mechanisms by, for example, chemical reduction in vitro[27,45]. However, as the overexpression of target proteins leads to higher levels compared with the endogenous system, elevated compound concentrations were required for binding. Despite this limitation, the methodology turned out to be robust, demonstrating that HpGroEL was bound and inhibited by **Metro** and the ether analogue **Metro-P3** in their activated forms to a similar extent. By contrast, the binding and turnover of HpTpx were notably more affected by **Metro-P3**, providing a first glimpse into the molecular reason for the diverging activity of the ether compared with the unmodified alcohol compounds. Thus, the more pronounced effect on Tpx led us to focus subsequent mechanistic investigations on this target. Given the known induction of oxidative stress within *H. pylori* cells by nitroimidazoles after reductive activation, the simultaneous inhibition of the essential oxidative stress response enzyme HpTpx results in a substantially decreased efficacy in detoxifying reactive radical species, which consequently triggers the accumulation of oxidative stress and, ultimately, cell death.

The molecular origin for the more selective binding of ether analogues to this target was elucidated by crystallography. Although **Metro** and **Metro-P3** form the same irreversible thioether bond with active site C$_P$ of HpTpx (Fig. 4b,c), both structures revealed different binding modes to HpTpx (Fig. 4e). Although the HpTpx complex with **Metro** depicts substantial structural rearrangements in helix α1, such conformational changes are absent in HpTpx with **Metro-P3**. The modelling of **Metro** into the HpTpx-**Metro-P3*** co-structure demonstrated a precise fit within the substrate binding pocket (Supplementary Fig. 18f). However, the less pronounced interactions of **Metro** with important enzyme residues and the absence of

the propargyl ether group, crucial for the prominent helix–dipole interaction in HpTpx-**Metro-P3***, explain its reduced binding affinity to HpTpx. Conversely, **Metro-P3** failed to be accommodated into the binding pocket of **Metro** (Supplementary Fig. 18g). Interestingly, prominent hydrogen bonds, formed between the generated amine groups of **Metro** with the carbonyl oxygen of C$_P$ and the amine group of **Metro-P3** with carbonyl oxygen of Ser54, provide a rationale for the inhibition of HpTpx$^{red}$ solely by the reduced ligands, explaining why the nitro-prodrugs cannot act as inhibitors of HpTpx. Overall, these results highlight a dual mode of action, which was so far overlooked for metronidazole and is triggered by the novel ether analogues.

As an outlook, the simultaneous inhibition of essential oxidative stress response enzymes by the 5-nitroimidazole ether analogues causes a perfect storm of damage and attenuated rescue (Fig. 5c). As the general antioxidant glutathione (GSH)-Grx system is absent in *S. aureus*, *M. tuberculosis* and *H. pylori*, the functional Trx system is essential for these strains to survive under oxidative stress and, thus, provides an opportunity to kill GSH-negative bacteria[46–52].

Although full safety and dosing studies will be required for any new molecule, the fact that 5-nitroimidazole ethers are derived from a clinically approved drug provides a strong foundation for preclinical development. Satisfyingly, crucial parameters such as toxicity, plasma stability and ADME properties were already suitable for most derivatives and do not require additional chemical optimization. With in vivo efficacy in a murine model, our hit compound **MF-01** is a promising candidate for further clinical testing and a valuable contribution towards a safer and more effective treatment of pathogenic *H. pylori* infections.

## Methods

The Supplementary Information gives additional detailed methods and protocols.

### Media, reagents and buffers for bacterial growth

WC anaerobe agar and brain heart infusion broth (BHI) were purchased from Oxoid; *H. pylori* selective supplement (Dent) (with vancomycin 5.0 mg, trimethoprim 2.5 mg cefsulodin 2.5 mg, amphotericin B 2.5 mg) was purchased from Oxoid. Penicillin–streptomycin (10,000 U ml$^{-1}$) was purchased from Gibco.

For the LB medium, we used peptone (10 g), NaCl (5 g) and yeast extract (5 g) in 1 l double-distilled water (ddH$_2$O), pH of 7.5; for the B medium, we used peptone (10 g), NaCl (5 g), yeast extract (5 g) and K$_2$PO$_4$ (1 g) in 1 l ddH$_2$O, pH of 7.5. For the BHI medium, we used 7.5 g brain infusion, 10 g peptone, 10 g heart infusion, 5 g NaCl, 2.5 g Na$_2$HPO$_4$ and 2 g glucose in 1 l ddH$_2$O, pH of 7.4.

For the AAM medium, we used BHI (18.5 g l$^{-1}$), yeast extract (5 g l$^{-1}$), trypticase soy broth (15 g l$^{-1}$), K$_2$HPO$_4$ (2.5 g l$^{-1}$), D-glucose (5 g l$^{-1}$), haemin (1 mg l$^{-1}$), Na$_2$CO$_3$ (0.4 g l$^{-1}$), cysteine hydrochloride (0.5 mg l$^{-1}$), menadione (5 mg l$^{-1}$) and 3% (v/v) FCS (heat inactivated).

### Bacterial strains and plasmids

**Bacterial strains.** *C. difficile* (DSM 27543), *Bacteroides caecimuris* (DSM 26085), *Muribaculum intestinale* (DSM 28989), *Enterococcus faecalis* (DSM 32036) and *Bifidobacterium animalis* (DSM 26074) were bought from Deutsche Sammlung von Mikroorganismen und Zellkulturen DSMZ. *H. pylori* strains (26695, PMSS1, SS1) were kindly provided by Prof. Markus Gerhard. All other commercially available strains were bought from American Type Culture Collection or Deutsche Sammlung von Mikroorganismen und Zellkulturen DSMZ. All bacteria classified within Bio/S2 are handled either under HeraSafe KS fume hoods (Thermo Fisher Scientific) or within an anaerobic chamber (Whitley DG250 anaerobic working station).

*E. coli* strains used within this work (BL21(DE3), Top10 and XL1Blue, DH5α, MM28) were cultivated in LB medium (37 °C, 200 rpm) and supplemented with respective antibiotics if indicated. The *S. aureus* strain (NCTC 8325) was cultivated aerobically in B medium (37 °C, 200 rpm).

*H. pylori* strains (26695, 26695 Δ*tpx*, PMSS1, SS1) were cultured on WC sheep blood (10%) agar plates (WC) that were supplemented with 1% *H. pylori* selective supplement (Dent) and incubated at 37 °C under microaerophilic (microaerophilic: 5% $CO_2$, 10% $O_2$ and 85% $N_2$) conditions. The broth cultures were prepared by inoculating BHI + 10% FCS medium with *H. pylori* colonies and then incubated for at least 24 h (37 °C, 140 rpm, microaerophilic) until the stationary phase was reached. For the growth curve determination, the bacterial growth was followed time-dependently via optical density at 600 nm ($OD_{600}$) measurements. The growth of *H. pylori* was additionally confirmed via Stuart's urease test (20 g $l^{-1}$ urea, 9.5 g $l^{-1}$ $Na_2HPO_4$, 9.1 g $l^{-1}$ $K_2HPO_4$, 0.1 g $l^{-1}$ yeast extract, 0.01 g $l^{-1}$ phenol red, pH of 6.8). *C. difficile* (DSM 27543) was cultured in BHI medium supplemented with 5 g $l^{-1}$ yeast extract, 0.1% (w/v) L-cysteine, 1 mg $l^{-1}$ resazurine in an anaerobic (anaerobic: 5% $H_2$, 10% $CO_2$ and 85% $N_2$) chamber (Whitley DG250 anaerobic working station). Anaerobic bacteria isolated from the murine gut including *B. caecimuris* (DSM 26085), *M. intestinale* (DSM 28989), *E. faecalis* (DSM 32036) and *B. animalis* (DSM 26074) were cultured in AAM medium in an anaerobic chamber.

**Plasmids.** Plasmids and their characteristics used for the methods in the following sections, including the protein purification of HpTpx, HpTpx point mutants, HpGroEL and Δ*tpx* knockout mutants, are summarized in Supplementary Table 8.

### Cell culture

Dulbecco's modified Eagle medium (DMEM; w, with: 4.5 g $l^{-1}$ glucose, w: L-glutamine, w/o, without: sodium pyruvate, w: 3.7 g $l^{-1}$ $NaHCO_3$) was purchased from PAN Biotech, high-glucose DMEM, FCS and PBS were purchased from Sigma-Aldrich. Trypsin–EDTA (0.25%) was purchased from Gibco. Cells were incubated at 37 °C and 5% $CO_2$ (Autoflow, Nuaire). The sterile workflow was performed under a laminar flow (HeraSafe KS, Thermo Scientific) equipped with a vacuum pump (BVC 21, Vacuumbrand).

HeLa cells (*Homo sapiens*, cervix carcinoma) and HepG2 (*Homo sapiens*, hepatocellular carcinoma) were grown in high-glucose DMEM medium supplemented with 10% (v/v) FCS (heat inactivated, Sigma) and 2 mM L-glutamine (Sigma). Gastric adenocarcinoma cells were cultured in a cell culture flask using DMEM supplemented with 10% (v/v) FCS and 1% (v/v) penicillin–streptomycin (10,000 U $ml^{-1}$) at 37 °C (5% $CO_2$). For determining the number of cells, the suspension was mixed with Trypan blue (1:1), and cells were counted in a Neubauer chamber under a Primo Vert microscope (Zeiss). Cell culture was checked daily, and polymerase chain reaction (PCR) testing for *Mycoplasma* infection was conducted before each experiment.

**MIC assay.** For MIC assays in *H. pylori* (26695, PMSS1), overnight cultures were inoculated from WC-Dent plates and grown overnight in BHI + 10% FCS (37 °C, 140 rpm, microaerophilic conditions, 5% $CO_2$, 10% $O_2$ and 85% $N_2$) and *H. pylori* growth was confirmed via Stuart's urease test. After measurement of $OD_{600}$, bacterial cultures were adapted to 0.5 McFarland units (0.5 McFarland = $OD_{600}$) of a solution comprising 99.5 ml 0.36 N $H_2SO_4$ + 0.5 ml 0.048 M $BaCl_2 \cdot 2H_2O$, $OD_{600}$ = 0.15; photometer blanked against air (Biophotometer Plus, Eppendorf). Bacterial suspensions were then diluted 1:100 in BHI + 10% FCS media and 50 µl were added to a sterile 96-well microtiter plate containing twofold serial dilutions of the compound in media (50 µl) in technical triplicates. After incubation for 96 h (37 °C, 140 rpm, microaerophilic conditions), the dilution series of each compound was analysed for microbial growth either by eye or by $OD_{600}$ readout with a Tecan reader (Promega GlowMax). The lowest compound concentration where no bacterial growth occurs was defined as the respective MIC value.

In all MIC experiments, a growth control containing DMSO and a sterile control containing medium only were included. Metronidazole was used as a control compound with known MIC in each biological independent experiment to ensure comparability of results. Experiments were performed in biological triplicates.

**Redox potential measurements.** Cyclic voltammetry measurements were carried out with a BioLogicSP200 potentiostat with EC-Lab software. Glassy carbon disk electrodes (3 mm diameter; PalmSens) were used as working and counter electrodes. Ag/AgCl (1 M KCl) was used as a reference electrode separated via a Vycor3535 frit (Advanced-Glass and Ceramics). Measurements were performed in a five-neck glass cell under argon atmosphere with respective nitroimidazole compounds ($c_{final}$ = 1 mM) in deoxygenated SSC buffer[53] (150 mM NaCl, 15 mM sodium citrate, pH of 7.0) with a scan rate of 100 mV $s^{-1}$, unless noted otherwise. Cyclic voltammograms are shown against $E$(Ag/AgCl) ($E$ being potential). Redox potentials are reported with reference to the normal hydrogen electrode (NHE): $E$(Ag/AgCl;1 M KCl) = 236 mV versus NHE.

**Resistance development assays.** Resistance development analysis is based on a procedure previously published by Ling et al.[54] as well as a procedure published by Hamamoto et al.[55]

For single-step FoR, overnight cultures of *H. pylori* 26695 were diluted into BHI + 10% FCS (500 µl) to a density of $10^5$ bacteria per millilitre and incubated at 37 °C, 100 rpm, microaerophilic conditions, 5% $CO_2$, 10% $O_2$ and 85% $N_2$. Serial dilutions ($10^{-5}$, $10^{-6}$, $10^{-7}$, $10^{-8}$) of two independent cultures each (400 µl) were plated onto WC agar plates containing 4×, 6×, 8×, 12× and 16× MIC of metronidazole, **MF-01**, **Metro-P3** or DMSO as growth control. After 5 days of incubation at 37 °C, 100 rpm, microaerophilic conditions, 5% $CO_2$, 10% $O_2$ and 85% $N_2$, the FoR was calculated as the mean number of mutants divided by the mean number of colonies detected on the growth control. The limit of detection (LOD) for the FoR was defined as $-\ln(0.05)/N$, where $N$ is the total number of colonies represented on growth control plates (corresponding to the 95% upper Poisson confidence bound for zero observed events).

For resistance development by sequential passaging, overnight cultures of *H. pylori* 26695 were diluted 1:100 into BHI + 10% FCS (500 µl) containing various concentrations of metronidazole, **MF-01**, **Metro-P3** or DMSO as growth control. Bacteria were incubated at 37 °C, 140 rpm, microaerophilic conditions, 5% $CO_2$, 10% $O_2$ and 85% $N_2$ and passaged in 72-h intervals in the presence of metronidazole, **MF-01** or **Metro-P3** at different concentrations (0.25×, 0.5×, 1×, 2× MIC). Cultures from the second highest concentration that allowed growth ($OD_{600}$ ≥0.9) were diluted 1:100 into fresh media (500 µl) containing different concentrations of the respective antimicrobial (0.25×, 0.5×, 1×, 2× MIC). If a shift in MIC levels was observed, concentrations of the respective antimicrobial were adjusted accordingly for subsequent passaging. This serial passaging was repeated for 9 days. The MIC shifts were calculated by dividing the respective daily MICs by the MIC of the DMSO-incubated culture.

**Gel-based ABPP in bacteria.** Bacterial overnight cultures were grown to early stationary phase (*H. pylori*: $OD_{600}$ ≈ 1.0, *S. pseudintermedius*: $OD_{600}$ ≈ 5.5, *S. schleiferi*: $OD_{600}$ ≈ 7.0). The cells were collected (6,000*g*, 15 min, r.t.) and washed with PBS once (15 ml, 6,000*g*, 15 min, 4 °C). The pellet was resuspended in PBS to obtain a bacterial culture of $OD_{600}$ ≈ 40. To 200 µl of $OD_{600}$ ≈ 40 culture, 2 µl **Metro-P1**, **Metro-P2** or **Metro-P3** probe (100× stock in DMSO, varying concentrations) or 1% (v/v) DMSO were added. Samples were incubated at 37 °C (*H. pylori*: 140 rpm, microaerophilic conditions; *S. pseudintermedius*, *S. schleiferi*: anaerobic conditions) for 2 h and subsequently collected by centrifugation (6,000*g*, 10 min, 4 °C). The supernatant was discarded, and the cell pellet was washed twice with 500 µl PBS and resuspended in 200 µl PBS* at 4 °C or in 200 µl 0.4% SDS–PBS. In the case of PBS, both cytosolic fraction (supernatant) and membrane fraction (pellet) were used for fluorescence click chemistry. Cytosolic fraction

was clicked directly, whereas for the membrane fraction, the supernatant was fully removed, and the pellet was washed once with 200 µl PBS before resuspension in 200 µl 0.4% SDS–PBS. Cells were either lysed by sonication (4× 20 s, 75% intensity with cooling breaks on ice) or *H. pylori* cell pellet was resuspended in radioimmunoprecipitation assay buffer with protease inhibitor (200 µl, 50 mM Tris-Cl, 150 mM NaCl, 1 mM EGTA, 1% Igepal, 0.25% sodium deoxycholate, pH of 7.4, addition of one tablet of protease inhibitor (Roche cOmplete Tablets Mini EDTA-free) per 10 ml radioimmunoprecipitation assay buffer), incubated on ice for 5 min followed by sonication for 10 min in a sonication bath (Bandelin Sonorex Super RK). Lysate was clarified by centrifugation (21,000$g$, 30 min, 4 °C in case of PBS, r.t. in case of 0.4% SDS–PBS as SDS precipitates at low temperatures). For the competitive labelling experiments, samples were pre-incubated with the respective concentrations of parent compound (metronidazole) at 37 °C (140 rpm, microaerophilic) for 2 h before addition of probe at defined concentrations.

The samples were subjected to click reaction by adding 1 µl rhodamine azide (10 mM in DMSO), 5 µl BTTAA (10 mM in DMSO), 2 µl CuSO$_4$ (50 mM in ddH$_2$O) and 2 µl NaAsc (100 mM in ddH$_2$O) to 90 µl of each sample and incubated for 1 h at r.t. The click reaction was quenched by adding 500 µl of cold acetone (−80 °C), vortexing and storing overnight at −80 °C. Samples were centrifuged (6,000$g$, 15 min, r.t.), and supernatant was aspirated. Cell pellet was resuspended in 65 µl PBS and 65 µl 2× SDS running buffer under mild ultrasound sonication (10 s, 10%, 5× cycle). Protein bands were separated using SDS–PAGE and visualized by fluorescence detection described in 'SDS–PAGE' section in the Supplementary Information.

**MS-based ABPP for in situ labelling.** *H. pylori* bacterial cultures (4× 250 ml) were grown to stationary phase (OD$_{600}$ ≈ 1.0, 90 h). Cells were collected (6,000$g$, 10 min, 4 °C) and washed with PBS once (15 ml, 6,000$g$, 10 min, 4 °C). The pellets were resuspended in PBS to obtain a bacterial culture of OD$_{600}$ ≈ 40. To 1 ml of OD$_{600}$ ≈ 40 culture, 10 µl probe (**Metro-P1**, **Metro-P2** or **Metro-P3**, 100× stock in DMSO, final concentration 1 µM) or 1% (v/v) DMSO were added and incubated at 37 °C (140 rpm, microaerophilic conditions) for 2 h and subsequently collected by centrifugation (6,000$g$, 10 min, 4 °C). The supernatant was removed, and the cell pellet was washed twice with 1 ml PBS before resuspension in 1 ml 0.4% SDS in PBS (*H. pylori*) or 200 µl in lysis buffer (200 µl, PBS, 0.5% SDS, 1% Triton X-100, *S. pseudintermedius*, *S. schleiferi*). *H. pylori* cells were lysed by sonication (4× 20 s, 75% intensity) with cooling breaks on ice; *S. pseudintermedius* and *S. schleiferi* cells were first resuspended by sonication (3× 10 s, 75% intensity with cooling breaks on ice) and then lysed using a Precellys homogenizer (3× 30 s, 30 s break). Lysate was clarified by centrifugation (20,000$g$, 30 min, r.t.). For competitive labelling experiments, 1 ml of OD$_{600}$ ≈ 40 *H. pylori* 26695 culture was pre-incubated with 100 µM metronidazole (100× excess, 100× stock in DMSO) at 37 °C (140 rpm, microaerophilic conditions) for 2 h before probes (1 µM) were added. The protein concentration of each sample was determined by bicinchoninic acid (BCA) assay described in 'BCA assay' section in the Supplementary Information.

The samples were subjected to click reaction by adding 5 µl biotin azide (final: 100 µM), 25 µl BTTAA (final: 500 µM), 10 µl CuSO$_4$ (final: 1 mM) and 10 µl NaAsc (final: 2 mM) to a total volume of 500 µl and incubated for 1 h at r.t. The click reaction was quenched by adding 2 ml cold acetone (−80 °C), vortexed and stored overnight at −80 °C. Precipitated proteins were pelletized (SLA3000, 10,000$g$, 15 min, 4 °C), and the supernatant was discarded. The protein pellet was washed with 0.5 ml cold methanol (−80 °C) and centrifuged (21,000$g$, 10 min, 4 °C) twice. Finally, protein pellets were resuspended in 500 µl 0.4% SDS–PBS by sonication (10 s, 10% intensity).

All following aqueous solutions were prepared using MS-grade water. Avidin beads (avidin-agarose from egg white, 1.1 mg ml$^{-1}$ in aqueous glycerol suspension, Sigma-Aldrich) were carefully resuspended by inversion on an Eppendorf tube wheel at 4 °C and transferred to an Eppendorf tube. The beads were washed with 0.4% SDS in PBS (3× 1 ml), centrifuged (400$g$, 3 min, r.t.) and filled to the original level with 0.4% SDS in PBS. In total, 50 µl of bead suspension was added to each 500 µl sample and incubated for 1 h at r.t. under continuous mixing. The samples were transferred quantitatively onto spin columns (BioEcho) and washed as follows: 0.4% SDS in PBS (3× 700 µl), 6 M urea in ddH$_2$O (2× 700 µl) and PBS (3× 700 µl) and transferred into new 1.5-ml LoBind tubes (Eppendorf) with 2× 100 µl X buffer (7 M urea, 2 M thiourea in 20 mM HEPES, pH of 7.5). Proteins on beads were subsequently reduced with Tris(2-carboxyethyl) phosphine (TCEP; final: 5 mM) for 1 h (37 °C, 600 rpm) and alkylated with iodoacetamide (IAA; final concentration 10 mM) for 30 min (r.t., 600 rpm). The reaction was quenched via addition of dithreithiol (DTT; final concentration 10 mM) for 30 min (r.t., 600 rpm). In total, 50 mM triethylammonium bicarbonate (TEAB; 600 µl) was added, and the pH value was checked to be ≥8.0. Trypsin (1.5 µl, 0.5 µg µl$^{-1}$ in 50 mM acetic acid, Promega) was added to digest the samples for 16 h (37 °C, 600 rpm) and quenched the next day by acidifying with formic acid (10 µl, final pH ≤3.0). The digested samples were centrifuged (13,000$g$, 3 min, r.t.) preceding the desalting step before MS measurement and analysis. Detailed methods are described in 'Desalting and filtration' and 'MS measurement and data analysis' sections in the Supplementary Information.

**IP-MS.** For IP-MS measurements, unmodified or modified proteins were diluted to a final concentration of 5–10 µM in MS-grade PBS and transferred to MS vials. For the labelling experiments of HpTpx with probes, HpTpx was diluted to a final concentration of 1 µM in MS-grade PBS. **Metro-P1**, **Metro-P2** and **Metro-P3** were added to a final concentration of 200 µM and incubated for 1 h (37 °C, 200 rpm) before the analysis described in the 'IP-MS measurement and data analysis' section in the Supplementary Information.

**ATPase activity assay.** A malachite green assay was performed to assess the ATPase activity of unmodified HpGroEL and modified HpGroEL protein with **Metro** (100%) or **Metro-P3** (71%)[56,57].

Before experiment conduction, malachite green reagent was prepared freshly by mixing 4.2% (w/v) ammonium molybdate in 4 M HCl and 0.045% (w/v) malachite green in H$_2$O in a 1:3 ratio and filtered through a 0.22-µm syringe filter. HpGroEL variants were diluted into the ATPase reaction buffer (50 mM Tris–HCl, 1 mM KCl/MgCl$_2$, 250 µM ATP, pH 7.5) to a final concentration of 1 µM. The reaction starts immediately after the addition of the protein. At specific time points (0 min, 5 min, 10 min, 20 min, 30 min, 45 min, 60 min and 120 min), 3× 40 µl of reaction mixture (technical replicates) were taken out and mixed with 160 µl of freshly prepared malachite green reagent. The absorbance was measured in a transparent 96-well plate at 620 nm using an Infinite M Nano Tecan 200Pro reader. As a negative (−) control experiments, HpGroEL was omitted from the reaction mixture. After baseline subtraction ((−) control), the initial ATPase activity rate was determined via simple linear regression (linear range, GraphPad Prism 10) of time-dependent absorbance measurements and normalized to unmodified HpGroEL activity (100%). All experiments were performed in seven biological replicates ($n_{bio}$ = 7) with technical triplicates per time point. Statistical significance of reduced ATPase activity was determined via ordinary one-way analysis of variance (ANOVA) with multiple comparisons (no correction) to unmodified HpGroEL protein.

**Peroxidase assay.** Peroxidase activity of the purified HpTpx protein was determined as previously described[34,58]. In a coupled assay with the Trx system (Trx/Trx reductase), rates of peroxide reduction were determined by indirectly monitoring NADPH oxidation as a decrease in absorbance at 340 nm. The assays were performed in 96-well plates in 100 µl reaction mixtures containing 200 nM trxR (recombinant

*E. coli* Trx reductase, abcam), 1 μM trx (recombinant *E. coli* Trx 1, abcam), 400 nM (un)modified HpTpx, 200 μM NADPH and 200 μM $H_2O_2$ in HEPES–NaOH (50 mM, pH OF 7.0). The reaction was initiated by addition of NADPH, and a decrease in absorbance at 340 nm was measured using an Infinite M Nano Tecan 200Pro reader after 5 min incubation time. As control experiments, Trx system, HpTpx and NADPH were sequentially omitted from the reaction mixtures. After the subtraction of baseline ((−) HpTpx), initial peroxide reduction rates were determined via simple linear regression (linear range, GraphPad Prism 9/10) of time-dependent absorbance measurements. For normalized activities, slopes were corrected by subtraction of baseline ((−) HpTpx) before normalization to unmodified HpTpx. The assay was performed in four (**Metro**) or five (**Metro-P3**) biologically independent experiments with three technical replicates per experiment ($n = 4$–5). The significance of reduced peroxidase activity was determined on unnormalized data via ordinary one-way ANOVA with multiple comparisons (no correction) of **Metro** or **Metro-P3** modified HpTpx protein to the unmodified wt HpTpx protein.

In peroxidase assays of unmodified HpTpx with and without the addition of compound (Supplementary Fig. 7), the protein was incubated with **Metro**, **Metro-P1**, **Metro-P2** and **Metro-P3** (final concentration 10 μM) for 30 min at r.t. before all other reagents were added, and absorbance at 340 nm was measured over time. The assay was performed in technical triplicates ($n_{tech} = 3$). Statistical significance was determined by ordinary one-way ANOVA with multiple comparisons (no correction).

**Binding site identification isoDTB workflow.** Binding site identification was performed with lysates obtained from ABPP MS-based labelling studies in *H. pylori* with **Metro-P3** (1 μM) with modifications to a previously published MS workflow using the isoDTB workflow procedure, where the digestion and enrichment steps were swapped[42,59].

In total, four samples of two biological replicates were prepared each as a light- or heavy-tagged version. For this, the protein concentration was adjusted to 1 mg ml⁻¹ in 0.4% (w/v) SDS in PBS (absolute protein amount per sample *H. pylori* 26695: 0.5 mg). A Click mix solution with heavy and light isoDTB (desthiobiotin azide) azide tags was prepared, where in total two tubes were mixed with TBTA (Tris[(1-benzyl-1*H*-1,2,3-triazol-4-yl)methyl]amine, 90 μl, 0.9 mg ml⁻¹ stock in 4:1 *t*BuOH/DMSO), TCEP (30 μl, 13 mg ml⁻¹ stock in ddH₂O) and CuSO₄ (30 μl, 50 mM stock in ddH₂O). A total of 30 μl of heavy or light isoDTB tag (5 mM stock in DMSO) was added to the tube. Click mix solutions (60 μl) were added to the **Metro-P3**-labelled *H. pylori* samples (2× heavy, 2× light) and incubated at r.t. for 1 h. After combining heavy and light samples of each biological replicate, cold acetone (800 μl, −20 °C) was added to both tubes, and samples were incubated at −20 °C for at least 2 h for full protein precipitation. The samples were centrifuged (13,000*g*, 4 °C, 10 min) and resulting protein pellet was washed two times with MeOH (500 μl, −20 °C) via mild sonication (10%, 5× cycle, 10 s) and centrifugation (13,000*g*, 4 °C, 10 min). The supernatant was removed, and the protein pellet was air-dried for 10 min. After dissolving the pellet in urea (300 μl, 8 M in 0.1 M TEAB buffer) via mild sonication (10%, 5× cycle, 10 s), the sample was centrifuged (13,000*g*, r.t., 3 min). For the reduction of disulfide bonds, DTT (15 μl, 31 mg ml⁻¹ in ddH₂O) was added to the samples and incubated at 37 °C for 45 min with constant shaking (850 rpm, Thermomixer; Eppendorf). IAA (15 μl, 74 mg ml⁻¹ stock in ddH₂O) was added for cysteine carbamidomethylation and incubated at 37 °C for 30 min while shaking (850 rpm, Thermomixer, Eppendorf). After the addition of DTT (15 μl, 31 mg ml⁻¹ stock in ddH₂O) to quench excess IAA, samples were shaken at 37 °C for further 30 min (850 rpm, Thermomixer, Eppendorf). In total, 900 μl of 0.1 M TEAB was added to each tube, and samples were digested with 20 μl of trypsin (trypsin:protein ratio of 1:100, 0.5 μg μl⁻¹ in 50 mM acetic acid; Promega) overnight while shaking (37 °C, 220 rpm, Incubator Shaker; Eppendorf).

Subsequently, the peptide solution was diluted 1:1 using 0.2% (v/v) NP-40 in PBS, and isoDTB-conjugated peptides were enriched on high-capacity streptavidin agarose beads (50 μl slurry, Fisher Scientific, 10733315) in 0.1% (v/v) NP-40 in PBS for 1 h at r.t while rotating (disc rotator). The suspension was centrifuged (1,000*g*, 2 min, r.t.), and the supernatant was removed. NP-40 (600 μl, 0.1% in PBS) was added to each sample and transferred to centrifugation columns (Fisher Scientific Pierce) to perform washing by gravity flow. Each sample was washed with NP-40 (1× 600 μl, 0.1% in PBS), with PBS (3× 600 μl) and ddH₂O (3× 600 μl). Peptides were eluted into tubes (Eppendorf) with 1× 200 μl and 2× 100 μl elution buffer (0.1% TFA in 50% aqueous MeCN) followed by a final centrifugation step (3,000*g*, 3 min, r.t.). The solvent was removed in a speed vac (5 h, 30 °C, Concentrator Plus, Eppendorf), and the resulting dried peptides were stored at −20 °C until analysis, described in 'Binding site identification isoDTB analysis' section in the Supplementary Information.

**Crystallization methods.** See 'Recombinant protein overexpression from *E. coli*' and 'Purification of recombinantly expressed proteins from *E. coli*' sections in the Supplementary Information for detailed protein expression conditions. For co-crystallization experiments, HpTpx C94A (HpTpx_$C_R$A) mutant was recombinantly expressed in the presence of **Metro** or **Metro-P3** as previously described.

The crystallization of HpTpx^red and its complexes, HpTpx_$C_R$A:**Metro*** (co-crystal structure of HpTpx C94A mutant with **Metro** in its amine form (*)) and HpTpx_$C_R$A:**Metro-P3*** (co-crystal structure of HpTpx C94A mutant with **Metro-P3** in its amine form (*)), was performed using the sitting drop vapour diffusion method at 20 °C. Protein solutions (20–30 mg ml⁻¹) were mixed with reservoir solutions in ratios of 1:1, 2:1 or 3:1, forming drops with a maximum volume of 0.4 μl. Specific conditions for crystal growth and data analysis are described in 'Crystallography' section in the Supplementary Information.

**In vitro ADME studies.** In vitro ADME studies comprised the assessment of plasma stability, metabolic stability assay as well as the plasma protein binding of 5-nitroimidazole compounds. These assays were conducted with selected compounds **Metro-P1**, **Metro-P3**, **MF-01**, **MF-02**, **MF-03** and **MF-07** as described previously[60]. Respective samples were analysed as described in 'HPLC–MS/MS analysis for ADME and PK studies' section in the Supplementary Information.

**PK studies.** For pharmacokinetic (PK) experiments, outbred male CD-1 mice (Charles River), 4 weeks old, were used. Compounds **Metro-P3**, **MF-01**, **MF-02** and **MF-03** were administered at 10 mg kg⁻¹ perorally (by using a peroral gavage; $n = 2$ per compound). Compounds were dissolved in a formulation containing 10% DMSO and 90% PBS. At the time points 0.5, 1 and 3 post administration, up to 25 μl of blood were collected from the lateral tail vein as well as spontaneous urine and faeces. At 5 h post administration, mice were killed to collect blood from the heart. Whole blood was collected into Eppendorf tubes coated with 0.5 M EDTA and immediately spun down at 15,870*g* for 10 min at 4 °C. Then, plasma was transferred into a new Eppendorf tube. Plasma, urine and faeces samples were stored at −80 °C until analysis. Moreover, focused PK studies were conducted for metronidazol, **Metro-P3** and **MF-03** at a dose of 7.1 mg kg⁻¹ perorally each. During the focused PK studies, $n = 3$ CD-1 mice were euthanized per time point ($t = 0.5, 1, 2, 4, 8$ and 24 h post administration) to conduct a stomach and a small intestine lavage, to collect whole blood from the heart, urine, faeces, stomach, small intestine and colon. For the stomach lavage, the stomach was taken out, lavaged with initially 500 μl isotonic sodium chloride solution, followed by 3 ml isotonic sodium chloride solution and finally 500 μl isotonic sodium chloride solution. For the small intestine lavage, the small intestine was taken out and lavaged with 1 ml isotonic sodium chloride solution. The stomachs, small intestines and the colons were put into 3 ml isotonic

sodium chloride solution each and homogenized using a Polytron tissue homogenizer at 12,000 rpm.

All PK samples were analysed via high-performance liquid chromatography (HPLC)–MS/MS using an Agilent 1290 Infinity II HPLC system coupled to an AB Sciex QTrap 6500plus or an AB Sciex QTrap 7500 mass spectrometer. First, a calibration curve was prepared by spiking different concentrations of metronidazole, **Metro-P3**, **MF-01**, **MF-02** and **MF-03** into mouse plasma, mouse urine, mouse faeces, isotonic sodium chloride solution, homogenized stomach, homogenized small intestine or homogenized colon from CD-1 mice. Caffeine was used as an internal standard. In addition, quality control samples (QCs) were prepared for metronidazole, **Metro-P3**, **MF-01**, **MF-02** and **MF-03** in the same matrices. For metronidazole, **Metro-P3**, **MF-01**, **MF-02** and **MF-03**, the same extraction procedure was used: 7.5 µl of a plasma sample (calibration samples, QCs or PK samples) were extracted with 37.5 µl of a 50:50 mixture of acetonitrile and methanol containing 12.5 ng ml$^{-1}$ of caffeine as internal standard for 15 min at 2,000 rpm on an Eppendorf MixMate vortex mixer. Then, samples were spun down at 16,000$g$ for 10 min. Supernatants were transferred to standard HPLC glass vials. For urine samples, 15 µl of a sample (calibration samples, QCs or PK samples) were extracted with 35 µl of a 50:50 mixture of acetonitrile and methanol containing 12.5 ng ml$^{-1}$ of caffeine as internal standard for 15 min at 2,000 rpm on an Eppendorf MixMate vortex mixer. Then samples were spun down at 16,000$g$ for 10 min. Supernatants were transferred to standard HPLC glass vials. For faeces (calibration samples, QCs or PK samples), equal amounts of a faeces portion were diluted with 200 µl water containing 10% MeOH and put on a ultrasonic batch at 35 kHz for 2× 20 min. Then, 15 µl of the sample was extracted with 25 µl of a 50:50 mixture of acetonitrile and methanol containing 12.5 ng ml$^{-1}$ of caffeine as internal standard for 5 min at 2,000 rpm on an Eppendorf MixMate vortex mixer. Then, samples were spun down at 16,000$g$ for 10 min. Supernatants were transferred to standard HPLC glass vials. For stomach lavage, stomach, small intestine, colon and small intestine lavage (calibration samples and QCs (of the respective matrix) or PK samples), 50 µl of a sample was extracted with 50 µl of a 50:50 mixture of acetonitrile and methanol containing 12.5 ng ml$^{-1}$ of caffeine as internal standard for 15 min at 800 rpm on an Eppendorf MixMate vortex mixer. Then samples were spun down at 6,000$g$ for 20 min. Supernatants were transferred to Greiner V-bottom plates and sealed with a mat.

**In vivo efficacy studies in murine model.** Female C57BL/6 wild-type mice (Envigo) were housed under specific pathogen-free conditions and were fed ad libitum. The 6–8-week-old mice were infected with *H. pylori* SS1 by oral gavage with $2 × 10^8$ bacteria resuspended in 200 µl BHI with 20% FCS. Mice were infected twice with *H. pylori* at a time interval of 2 days. After a stable infection was established (6–8 weeks), mice were treated with different antibiotic triple therapies by oral gavage twice a day for 7 days: metronidazole or **Metro-P3** (14.2 mg kg$^{-1}$ day$^{-1}$) or **Metro-P3** (0.02 × 14.2 mg kg$^{-1}$ day$^{-1}$ = 0.30 mg kg$^{-1}$ day$^{-1}$) or **MF-01** (0.02 × 14.2 mg kg$^{-1}$ day$^{-1}$ = 0.30 mg kg$^{-1}$ day$^{-1}$) or metronidazole (0.02 × 14.2 mg kg$^{-1}$ day$^{-1}$ = 0.30 mg kg$^{-1}$ day$^{-1}$) with clarithromycin (7.15 mg kg$^{-1}$ day$^{-1}$) or clarithromycin only (control group, 7.15 mg kg$^{-1}$ day$^{-1}$). At 1 h before antibiotic treatment, mice were administered proton pump inhibitor omeprazol (400 µmol kg$^{-1}$ day$^{-1}$). One group was not treated with antibiotics as a positive colonization control group for CFU determinations. At 2 weeks after antibiotic treatment, mice were killed by cervical dislocation, and colonization was analysed by plating serial dilutions of stomach tissue. For this, a longitudinal piece of stomach was weighed and homogenized in 1 ml BHI + 20% FCS with a bead mill (Precellys 24, 5,000 rpm, 2× 45 s, 5 s breaks). Serial dilutions (*H. pylori* SS1: 1:10$^2$, 1:10$^3$, 1:10$^4$) in BHI + 20% FCS were prepared and plated on WC-Dent blood agar plates supplemented with bacitracin (200 µg ml$^{-1}$), nalidixic acid (10 µg ml$^{-1}$) and polymyxin B (3 µg ml$^{-1}$). After 5 days of culture, CFUs were counted

and optionally expanded on WC-Dent plates for 2 days and then frozen for later use. All experiments were approved by the Bavarian Government (Regierung von Oberbayern, ROB-55.2-2532.Vet_02-23-90) and conducted in compliance with European guidelines for the care and use of laboratory animals.

For in vivo efficacy studies in C57BL/6 mice infected with *H. pylori* SS1 strain were treated with different regimens: no treatment ($n = 7$), clarithromycin only ($n = 6$), triple therapy with low dose of **Metro-P3** (0.02× **Metro**; $n = 4$) or **MF-01** (0.02× **Metro**; $n = 5$), standard triple therapy with **Metro** ($n = 6$) or **Metro-P3** (1× **Metro**, $n = 5$). Statistical significance was determined by non-parametric Kruskal–Wallis test with multiple comparisons (no correction) to control column (no treatment). The LOD (CFU mg$^{-1}$) was defined as the equivalent of a single colony on the lowest dilution plated and calculated as LOD = $n × (DF/V_{plate}) × (V_{hom}/m_{proc})$, where DF is the dilution factor, $V_{plate}$ is the plated volume, $V_{hom}$ is the homogenate volume and $m_{proc}$ is the mass of processed tissue.

## Microbiome analysis

**DNA extraction.** DNA extractions from collected faecal samples were adapted from Turnbaugh et al.[61] (isoamyl alcohol extraction) and from Zoetendal et al.[62] (ethanol precipitation). In brief, an aliquot (≤200 mg) of each sample was suspended, while frozen, in a solution containing 500 µl of freshly prepared extraction buffer (200 mM Tris, 200 mM NaCl, 20 mM EDTA, pH of 8.0), 200 µl of 20% SDS and 500 µl of a mixture of phenol:chloroform:isoamyl alcohol (25:24:1) in 2 ml VK05 Precellys tubes containing 0.5 mm glass beads (Bertin Technologies). Next, a Bertin 24 homogenizer (Bertin Technologies) was run at 5,000 rpm (2 cycles of 45 s with a pause of 5 s in between) to lyse the microbial cells. This was followed by extraction with the isoamyl alcohol extraction method and precipitation with 3 M sodium acetate (pH of 5.2) and cold ethanol (99%, molecular grade stored at −20 °C). DNA pellets at the end of the protocol were dissolved in 200 µl TE buffer (pH of 8.0, heated to 70 °C).

**PCR and library preparation.** To target bacterial DNA, the V3/V4 region of the 16S rRNA gene was amplified in 25 cycles from 1–2 µl of aliquoted working stocks of DNA using primers 341F-ovh and 785R-ovh (Klindworth et al.[63]). AMPure XP magnetic beads (Beckmann Coulter Life Sciences) were used for purification of the amplicons according to Illumina's 16S Metagenomic Sequencing Library Preparation guide. Samples were indexed with Nextera XT indices (Illumina) in a paired fashion for eight cycles of PCR, followed by a second purification with AMPure XP beads. Indexed samples were pooled in an equimolar amount (4 nM), adjusted to a final concentration of 4 pM and sequenced on a MiSeq system (Illumina) in a paired-end reaction of 600 cycles using the MiSeq reagent kit v3 (Illumina). A 20% (v/v) spike-in of the PhiX standard library at 4 pM was additionally included. As a control to check for artifacts, a single negative control (PCR with nuclease-free water as template DNA) as well as a positive control using a mock community (ZymoBIOMICS, number D6300) were included throughout each sequencing run.

Primers

| Name | Sequence 5' > 3' |
|---|---|
| 341F-ovh | <u>TCGTCGGCAGCGTCAGATGTGTATAAGAGACAG</u>CCTACGGGN GGCWGCAG |
| 785R-ovh | <u>GTCTCGTGGGCTCGGAGATGTGTATAAGAGACAG</u>GACTACHV GGGTATCTAATCC |

Underlined portions indicate the Illumina-specific adapter overhang sequences (16S Metagenomic Sequencing Library Preparation, Illumina). Non-underlined portions target the 16S rRNA V3/V4 region, as designed and reported in Klindworth et al.[63].

**Bioinformatics analysis.** The output was obtained from the sequencing in a compressed and demultiplexed fastq.gz file format and was processed using the QIIME2 platform (v.2024.5) (Bolyen et al.[64]). First, overhang adaptor and primer sequences were trimmed from the paired-end reads using Cutadapt plugin (Martin[65]). These were then filtered, denoised and checked for chimeras before ASV assignment using the DADA2 plugin (Callahan et al.[66]). Taxonomic classification was assigned in the QIIME2 pipeline using the classify-sklearn algorithm under feature-classifier plugin against a pretrained classifier based on the Greengenes2 2024.9 release (McDonald et al.[67]). Each sample was rarefied to 12,409 sequences (size of the smallest sample) for further downstream analyses such as calculation of richness, diversity and community compositional indices and matrices. Statistical tests were performed in R (R Foundation for Statistical Computing, Vienna, Austria; https://www.r-project.org/), and statistical significance was determined by the non-parametric Kruskal–Wallis test with multiple comparisons (no correction) to control column in GraphPad Prism 10.

**Ethics statement.** The animal studies were conducted in accordance with the recommendations of the European Community (Directive 2010/63/EU, 1 January 2013). All animal procedures were performed in strict accordance with the German regulations of the Society for Laboratory Animal Science and the European Health Law of the Federation of Laboratory Animal Science Associations. Animals were excluded from further analysis if killing was necessary according to the humane endpoints established by the ethical board. All experiments were approved by the ethical board of the Niedersächsisches Landesamt für Verbraucherschutz und Lebensmittelsicherheit, Oldenburg, or by the Bavarian government (Regierung von Oberbayern, permit number ROB-55.2-2532.Vet_02-23-90). Animals were kept in individually ventilated cages with a 10 h–14 h dark–light cycle and had access to food and water ad libitum as well as additional nesting material. The animals were housed in individually ventilated cages in an animal facility with a room temperature (r.t.) of 21–22 °C, a 12 h light–12 h dark cycle (lights on at 07:00, off at 19:00) and a relative humidity in the range of 45–65%.

**Reporting summary**

Further information on research design is available in the Nature Portfolio Reporting Summary linked to this article.

## Data availability

The data that support the findings of this study are available within the Article and its Supplementary Information. Crystal structures have been deposited in the RCSB Protein Data Bank (PDB IDs: 2XPD, 9F5V, 3ZRD, 9F64, 9F65). The MS proteomics data have been deposited to the ProteomeXchange Consortium[68] via the PRIDE[69] partner repository with the dataset identifier PXD051773 (*H. pylori*) and PXD067473 (*S. epidermidis*, *S. pseudintermedius*). The 16S sequencing data have been deposited to ENA (PRJEB107125). Source data are provided with this paper.

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

## Acknowledgements

We thank P. Wunderlich, E. Geiger, R. Priller, Z. Chen, H. Kramer, M. Wefelmeyer, O. Omelko and C. Scherf for helping with synthesis, cloning and biological activity assays. We thank T. Burrell, M. Skerhut, K. Mink, A. König, M. Wolff, K. Bäuml, A. Ahlers, J. Schreiber, S. Wachowsky and J. Wolf for their technical assistance. We thank the staff of the beamline X06SA at the Paul-Scherrer-Institute,

Swiss Light Source (SLS), Villigen, Switzerland, and the staff of the beamline ID30B, European Synchrotron Radiation Facility (ESRF), Grenoble, France, for assistance during data collection. We thank the biopharmaceutical company *Ondek* for providing the plasmid pOND708 for generating the Δ*tpx* mutant. This project was funded by the European Union, ERC, breakingBAC, grant number 101096911, and received funding support from the German Research Foundation CRC1371 project number 395357507 TP-P16 and was supported in part by the Merck Future Insight Prize awarded by Merck, Darmstadt, Germany. M.Ge. and R.M.L. were funded by the Bavarian bayresq.net project Heliresis. Additional financial support was provided by the DFG, grant number SFB 1309-325871075 (M.Gr.). M.K.F. acknowledges a Kekulé stipend from 'Fond der Chemischen Industrie'. K.R. receives funding from the German Center for Infection Research (DZIF, grant number TTU 09.719).

## Author contributions

Conceptualization: M.K.F. and S.A.S. Methodology: M.K.F., M.S.I.P., R.G., S.F., K.R., D.P., T.R., C.M. and M.H. Investigation: M.K.F., M.S.I.P., R.G., S.F., K.R., D.P., C.M., M.H. and D.S. Crystallography: M.Gr. Visualization: M.K.F., M.S.I.P. and M.Gr. Funding acquisition: S.A.S., M.Ge., M.Gr. and K.R. Project administration: M.K.F. and S.A.S. Supervision: S.A.S., M.Ge., C.R.H., R.M.L., M.Gr. and V.F. Writing—original draft: M.K.F., S.A.S. and M.Gr. Writing—proofreading and editing: M.K.F., M.S.I.P., R.G., S.F., K.R., V.F., D.P., T.R., C.M., M.H., C.R.H., R.M.L., M.Ge., M.Gr. and S.A.S.

## Funding

## Competing interests

SAS is co-founder of smartbax limited. The other authors declare no competing interests.

## Additional information

**Correspondence and requests for materials** should be addressed to Stephan A. Sieber.

[1]TUM School of Natural Sciences, Department Biosciences, Chair of Organic Chemistry II, Center for Functional Protein Assemblies, Technical University Munich, Garching, Germany. [2]TUM School of Medicine and Health, Department Preclinical Medicine, Institute of Medical Microbiology, Immunology and Hygiene, Technical University of Munich, Munich, Germany. [3]German Center for Infection Research, Partner Site Munich, Munich, Germany. [4]Department of Pediatric Gastroenterology, Shanghai Children's Medical Center, School of Medicine, Shanghai Jiao Tong University, Shanghai, China. [5]Department of Chemical Biology, Helmholtz Centre for Infection Research, Braunschweig, Germany. [6]German Center for Infection Research, Partner Site Hannover-Braunschweig, Braunschweig, Germany. [7]TUM School of Natural Sciences, Department of Chemistry, Catalysis Research Center, Technical University of Munich, Garching, Germany. [8]Faculty of Chemistry and Pharmacy, University of Regensburg, Regensburg, Germany. [9]TUM School of Natural Sciences, Department Biosciences, Chair of Biochemistry, Center for Functional Protein Assemblies, Technical University Munich, Garching, Germany. [10]These authors contributed equally: Michaela K. Fiedler, Marianne S. I. Pandler. ✉e-mail: stephan.sieber@tum.de

# Reporting Summary

## Statistics

For all statistical analyses, confirm that the following items are present in the figure legend, table legend, main text, or Methods section.

| n/a | Confirmed | |
|---|---|---|
| ☐ | ☒ | The exact sample size (*n*) for each experimental group/condition, given as a discrete number and unit of measurement |
| ☐ | ☒ | A statement on whether measurements were taken from distinct samples or whether the same sample was measured repeatedly |
| ☐ | ☒ | The statistical test(s) used AND whether they are one- or two-sided *Only common tests should be described solely by name; describe more complex techniques in the Methods section.* |
| ☐ | ☒ | A description of all covariates tested |
| ☐ | ☒ | A description of any assumptions or corrections, such as tests of normality and adjustment for multiple comparisons |
| ☐ | ☒ | A full description of the statistical parameters including central tendency (e.g. means) or other basic estimates (e.g. regression coefficient) AND variation (e.g. standard deviation) or associated estimates of uncertainty (e.g. confidence intervals) |
| ☐ | ☒ | For null hypothesis testing, the test statistic (e.g. *F*, *t*, *r*) with confidence intervals, effect sizes, degrees of freedom and *P* value noted *Give P values as exact values whenever suitable.* |
| ☒ | ☐ | For Bayesian analysis, information on the choice of priors and Markov chain Monte Carlo settings |
| ☒ | ☐ | For hierarchical and complex designs, identification of the appropriate level for tests and full reporting of outcomes |
| ☒ | ☐ | Estimates of effect sizes (e.g. Cohen's *d*, Pearson's *r*), indicating how they were calculated |

*Our web collection on statistics for biologists contains articles on many of the points above.*

## Software and code

Policy information about availability of computer code

| Data collection | no code was used |
|---|---|
| Data analysis | GraphPad Prism 10, Perseus 2.0.10.0 or 2.0.11.0, ImageJ, ChemDraw, MestReNova, FragPipe, MSFragger, MaxQuant 1.6.2.10 |

For manuscripts utilizing custom algorithms or software that are central to the research but not yet described in published literature, software must be made available to editors and reviewers. We strongly encourage code deposition in a community repository (e.g. GitHub). See the Nature Portfolio guidelines for submitting code & software for further information.

## Data

Policy information about availability of data

All manuscripts must include a data availability statement. This statement should provide the following information, where applicable:

- Accession codes, unique identifiers, or web links for publicly available datasets
- A description of any restrictions on data availability
- For clinical datasets or third party data, please ensure that the statement adheres to our policy

Crystal structures have been deposited in the RCSB Protein Data Bank (PDB IDs: 2XPD, 9F5V, 3ZRD, 9F64, 9F65). The mass spectrometry proteomics data have been deposited to the ProteomeXchange Consortium via the PRIDE partner repository with the dataset identifier PXD051773 (H. pylori) and PXD067473 (S. epidermidis, S. pseudintermedius). 16S sequencing data have been deposited to ENA (PRJEB107125).

# Research involving human participants, their data, or biological material

Policy information about studies with <u>human participants or human data</u>. See also policy information about <u>sex, gender (identity/presentation), and sexual orientation</u> and <u>race, ethnicity and racism</u>.

| | |
|---|---|
| Reporting on sex and gender | no human participant involved in the present study |
| Reporting on race, ethnicity, or other socially relevant groupings | no human participant involved in the present study |
| Population characteristics | no human participant involved in the present study |
| Recruitment | no human participant involved in the present study |
| Ethics oversight | no human participant involved in the present study |

Note that full information on the approval of the study protocol must also be provided in the manuscript.

# Field-specific reporting

Please select the one below that is the best fit for your research. If you are not sure, read the appropriate sections before making your selection.

☒ Life sciences ☐ Behavioural & social sciences ☐ Ecological, evolutionary & environmental sciences

For a reference copy of the document with all sections, see <u>nature.com/documents/nr-reporting-summary-flat.pdf</u>

# Life sciences study design

All studies must disclose on these points even when the disclosure is negative.

| | |
|---|---|
| Sample size | Sample sizes were selected based on standard practice in the field. For in vivo mouse studies a statistician determined the sample size during the ethical approval process. |
| Data exclusions | No data was excluded. |
| Replication | Exact replicate numbers are indicated individually for each experiment in the manuscript. All attempts at replication were successful. |
| Randomization | Randomization was not performed/necessary. |
| Blinding | Blinding was not performed/necessary. |

# Reporting for specific materials, systems and methods

We require information from authors about some types of materials, experimental systems and methods used in many studies. Here, indicate whether each material, system or method listed is relevant to your study. If you are not sure if a list item applies to your research, read the appropriate section before selecting a response.

## Materials & experimental systems

| n/a | Involved in the study |
|---|---|
| ☒ | ☐ Antibodies |
| ☐ | ☒ Eukaryotic cell lines |
| ☒ | ☐ Palaeontology and archaeology |
| ☐ | ☒ Animals and other organisms |
| ☒ | ☐ Clinical data |
| ☒ | ☐ Dual use research of concern |
| ☒ | ☐ Plants |

## Methods

| n/a | Involved in the study |
|---|---|
| ☒ | ☐ ChIP-seq |
| ☒ | ☐ Flow cytometry |
| ☒ | ☐ MRI-based neuroimaging |

# Eukaryotic cell lines

Policy information about <u>cell lines and Sex and Gender in Research</u>

| | |
|---|---|
| Cell line source(s) | HeLa, AGS and HepG2 were bought from ATCC or DSMZ |

| Authentication | was not authenticated |
|---|---|
| Mycoplasma contamination | cell lines were tested for mycoplasma contamination prior to experiments |
| Commonly misidentified lines<br>(See ICLAC register) | there was no commonly misidentified line used |

# Animals and other research organisms

Policy information about studies involving animals; ARRIVE guidelines recommended for reporting animal research, and Sex and Gender in Research

| Laboratory animals | male CD-1 mice (Charles River, Germany) and Female C57BL/6 wild-type mice (Envigo) |
|---|---|
| Wild animals | no wild animals used |
| Reporting on sex | sex was not considered in particular |
| Field-collected samples | study did not involve sample collection |
| Ethics oversight | The animal studies were conducted in accordance with the recommendations of the European Community (Directive 2010/63/EU, 1st January 2013). All animal procedures were performed in strict accordance with the German regulations of the Society for Laboratory Animal Science (GV-SOLAS) and the European Health Law of the Federation of Laboratory Animal Science Associations (FELASA). Animals were excluded from further analysis if sacrifice was necessary according to the humane endpoints established by the ethical board. All experiments were approved by the ethical board of the Niedersächsisches Landesamt für Verbraucherschutz und Lebensmittelsicherheit, Oldenburg, or by the Bavarian government (Regierung von Oberbayern, permit no. ROB-55.2-2532.Vet_02-23-90). Animals were kept in individually ventilated cages with a 10h/14h dark/light cycle and had access to food and water ad libitum as well as additional nesting material. The animals were housed in individually ventilated cages in an animal facility with a room temperature of 21–22 °C and a relative humidity in the range of 45–65%. |

Note that full information on the approval of the study protocol must also be provided in the manuscript.

# Plants

| Seed stocks | no plants were used in the present study |
|---|---|
| Novel plant genotypes | no plants were used in the present study |
| Authentication | no plants were used in the present study |

