## [Peer Review File · Nature Microbiology]

Metronidazole and ether derivatives target *Helicobacter pylori* via simultaneous stress induction and inhibition

Corresponding Author: Professor Stephan Sieber

Version 0:

Reviewer comments:

Reviewer #1

(Remarks to the Author)

Fiedler et al. have provided substantive updates to their manuscript, now titled "Dual-targeting 5-nitroimidazole ethers eradicate *H. pylori* by induction of stress and disability its adaptive response", that was previously reviewed for Nature. The revisions address key concerns raised by all reviewers and span all aspects of the paper, including additional in vitro and in vivo data. These updates increase the interpretability of the results and improve the manuscript. This work continues to be interesting and relevant for publication in Nature Microbiology. I will just re-raise a couple comments, though none rise to a level of concern at this point:

Figure 2, S6, S7, and S8 – The authors now provide a rationale for the focus on HpTpx and HpGroEL, which is helpful, and the totality of the data are consistent with their choice. However, I am still stuck on the possibility that there is more going on with some of these analogs suggesting additional targets could be explored to fully understand the activities of these molecules.

Figure 5C – The authors note in the figure legends that these are triple cocktail therapies, but it would be helpful to also capture this in the actual figure to avoid any confusion that these animals were not treated with Metro-P3, MF-01, or Standard Metro alone. Also, in the legends for this panel, the authors now note that the zero values are genuine (in response to a good catch by Reviewer #2). I am a little confused though: does this mean that the entirety of the stomach was plated and zero CFU were formed? If the entire organ was not plated, there should still be a limit of detection per gram based on the plated dilutions, correct?

Line 366 – Can the authors please add references for the work that previously synthesized these compounds (even if it was for other applications)?

Reviewer #2

(Remarks to the Author)

This revision partially addresses critiques, but major issues remain.

According to the paper, the most significant findings are that the metronidazole analogs are 50 fold more effective than metronidazole in clearing an *H. pylori* infection in a mouse model; and that the analogs have a dual mode of action, in contrast to metronidazole. These assertions do not seem to be supported by the evidence presented.

Mechanism of action.

Responding to a previous critique, the authors note in the rebuttal: "It is important for us to underline that the main impact from this study results from deciphering the diverging MoAs of metronidazole and 5-nitroimidazole ethers, as this might have not been clear in our previous version: While both molecules induce oxidative stress and inhibit GroEL, the ether derivatives address Tpx as an additional target to block the stress response which is deleterious after its stress induction. To emphasize this unprecedented dual mode of action more clearly, we changed the title of the manuscript accordingly and carefully revised the main text."

- This is a puzzling statement. As the authors report: "Interestingly, when we performed the same concentration-dependent protein expression experiments with HpTpx in *E. coli*, Metro-P3 resulted in a 3.6-fold higher HpTpx modification compared to Metro (Figure 3F)." This result directly contradicts the assertion that the analogs have an additional target. Rather, the analogs have a modest 3.6 fold improvement in engaging a second target, Tpx, over metronidazole.

Animal efficacy.

Fig. 3 and Supplemental Fig. S 24 describing the efficacy data are poorly designed, presented and are challenging to understand. The X axis labels ostensibly depict the compounds that decrease the pathogen burden shown in bar graphs, such

as "Metro-P3 1X". One assumes that this means that the decrease in pathogen shown in Metro-P3 1X bar is caused by Metro-P3 administration at some (not defined) 1X dose. This is not the case – Metro-P3 is actually part of a triple therapy regimen. The dose of Metro-P3, identity and dosage of those other compounds is not given in the figure legend. Since Metro-P3 and metronidazole are only administered as constituents of a triple therapy, drawing conclusions about their relative efficacy is uncertain. In a control, data with clarithromycin alone are presented. Metro-P3 and metronidazole should be similarly compared in the same experiment at the same doses as single compounds.

The authors specifically claim: "In the case of treatment with MF-01, already 0.30 mg/kg/day were sufficient for full *H. pylori* eradication, a dose at which Metro exhibited insufficient efficacy (Figure S24). This 50-fold reduction compared to Metro represents a major improvement in the treatment of *H. pylori* infections in vivo, significantly lowering drug exposure (Figure 5C)." However, they are not comparing MF-01 to a minimally efficacious dose of metronidazole, which remains to be determined. Rather, they compare MF-01 to a therapeutically recommended dose of metronidazole. Without knowing the minimally efficacious dose of metronidazole, it is impossible to draw any quantitative conclusions regarding the relative efficacy of these compounds. Note that experiment Fig. 5C is performed with Metro-P3, while S24 shows data for MF-01. This adds to the confusion.

Resistance.

In response to the critique, the authors perform an experiment with resistance development using sub-inhibitory levels of compounds and serial passage, and report that this resulted in the formation of round bodies. This is not informative. The standard test is frequency of resistance determination by plating pathogen on plates with various concentrations of drug starting at 4X MIC, and reporting the ratio of colonies grown/plated.

Recommendation.

The interesting observation of this study is the discovery of an additional target for the metronidazole series of compounds, Tpx. Focusing on the more compelling mechanistic studies and dropping inadequate microbiology and animal efficacy will improve the paper.

Reviewer #3

(Remarks to the Author)

In this extensive revision, the authors have responded to my initial review with substantial new data, including addition PK studies aimed at measuring exposure levels in relevant tissues, the synthesis and testing of additional control compounds, and repeat of proteomics studies with additional competitors. All of these studies add rigor to the results, which are surprising and significant, and should be of interest to the readers of Nature Microbiology. The response to the other reviewers comments and suggestions is similarly comprehensive. As such, I recommend publication of the revised manuscript and congratulate the authors on the work and their interesting findings.

Decision Letter:

1st October 2025

Dear Stephan,

Thank you for your patience while your manuscript "Dual-targeting 5-nitroimidazole ethers eradicate *H. pylori* by induction of stress and disabling its adaptive response" was under peer-review at Nature Microbiology. Thank you also for providing a revision plan. The manuscript has now been seen by 3 referees, whose comments you will find at the end of this email. As previously mentioned, while they find your work of interest, some important points are raised. We are very interested in the possibility of publishing your study in Nature Microbiology, but would like to consider your response to these concerns in the form of a revised manuscript before we make a final decision on publication.

In particular, in response to referee #1, please perform some text and figure edits to clarify their points. Regarding the concerns by referee #2, we agree that your arguments regarding an additional target versus improved engagement of the second target are valid. However, please caveat your points in the discussion. Please also clarify that you are using a triple therapy. Editorially, we feel the ethical concerns regarding additional animal experiments are valid. Please add additional in vitro experiments to bolster the resistance development section.

In summary, please provide text or figure edits for most points. Make sure you caveat any points clearly in the results or discussion sections. We will require you to address the resistance development point with in vitro work as described in your revision plan. The rest of the referees' reports are clear and the remaining issues should be straightforward to address.

If you have not done so already please begin to revise your manuscript so that it conforms to our Article format instructions at <http://www.nature.com/nmicrobiol/info/final-submission/>

The usual length limit for a Nature Microbiology Article is six display items (figures or tables) and 4,000 words. We have some flexibility, and can allow a revised manuscript at 4,500 words, but please consider this a firm upper limit. There is a trade-off of ~250 words per display item, so if you need more space, you could move a Figure or Table to Supplementary Information.

Some reduction could be achieved by focusing any introductory material and moving it to the start of your opening 'bold' paragraph, whose function is to outline the background to your work, describe in a sentence your new observations, and explain your main conclusions. The discussion should also be limited. Methods should be described in a separate section following the discussion, we do not place a word limit on Methods.

Nature Microbiology titles should give a sense of the main new findings of a manuscript, and should not contain punctuation. Please keep in mind that we strongly discourage active verbs in titles, and that they should ideally fit within 90 characters each (including spaces).

Please include a data availability statement as a separate section after Methods but before references, under the heading "Data Availability". This section should inform readers about the availability of the data used to support the conclusions of your study. This information includes accession codes to public repositories (data banks for protein, DNA or RNA sequences, microarray, proteomics data etc...), references to source data published alongside the paper, unique identifiers such as URLs to data repository entries, or data set DOIs, and any other statement about data availability. At a minimum, you should include the following statement: "The data that support the findings of this study are available from the corresponding author upon request", mentioning any restrictions on availability. If DOIs are provided, we also strongly encourage including these in the Reference list (authors, title, publisher (repository name), identifier, year). For more guidance on how to write this section please see: <http://www.nature.com/authors/policies/data/data-availability-statements-data-citations.pdf>

To improve the accessibility of your paper to readers from other research areas, please pay particular attention to the wording of the paper's opening bold paragraph, which serves both as an introduction and as a brief, non-technical summary in about 150 words. If, however, you require one or two extra sentences to explain your work clearly, please include them even if the paragraph is over-length as a result. The opening paragraph should not contain references. Because scientists from other sub-disciplines will be interested in your results and their implications, it is important to explain essential but specialised terms concisely. We suggest you show your summary paragraph to colleagues in other fields to uncover any problematic concepts.

If your paper is accepted for publication, we will edit your display items electronically so they conform to our house style and will reproduce clearly in print. If necessary, we will re-size figures to fit single or double column width. If your figures contain several parts, the parts should form a neat rectangle when assembled. Choosing the right electronic format at this stage will speed up the processing of your paper and give the best possible results in print. We would like the figures to be supplied as vector files - EPS, PDF, AI or postscript (PS) file formats (not raster or bitmap files), preferably generated with vector-graphics software (Adobe Illustrator for example). Please try to ensure that all figures are non-flattened and fully editable. All images should be at least 300 dpi resolution (when figures are scaled to approximately the size that they are to be printed at) and in RGB colour format. Please do not submit Jpeg or flattened TIFF files. Please see also 'Guidelines for Electronic Submission of Figures' at the end of this letter for further detail.

Figure legends must provide a brief description of the figure and the symbols used, within 350 words, including definitions of any error bars employed in the figures.

When submitting the revised version of your manuscript, please pay close attention to our [href="https://www.nature.com/nature-research/editorial-policies/image-integrity">Digital Image Integrity Guidelines](https://www.nature.com/nature-research/editorial-policies/image-integrity) and to the following points below:

EXTENDED DATA FIGURES

Please include a statement before the acknowledgements naming the author to whom correspondence and requests for materials should be addressed.

Finally, we require authors to include a statement of their individual contributions to the paper -- such as experimental work, project planning, data analysis, etc. -- immediately after the acknowledgements. The statement should be short, and refer to

authors by their initials. For details please see the Authorship section of our joint Editorial policies at http://www.nature.com/authors/editorial_policies/authorship.html

* include a point-by-point response to any editorial suggestions and to our referees. Please include your response to the editorial suggestions in your cover letter, and please upload your response to the referees as a separate document.

* ensure it complies with our format requirements for Letters as set out in our guide to authors at www.nature.com/nmicrobiol/info/gta/

* resubmit electronically if possible using the link below to access your home page:

Link Redacted

*This url links to your confidential homepage and associated information about manuscripts you may have submitted or be reviewing for us. If you wish to forward this e-mail to co-authors, please delete this link to your homepage first.

Please ensure that all correspondence is marked with your Nature Microbiology reference number in the subject line.

Nature Microbiology is committed to improving transparency in authorship. As part of our efforts in this direction, we are now requesting that all authors identified as 'corresponding author' on published papers create and link their Open Researcher and Contributor Identifier (ORCID) with their account on the Manuscript Tracking System (MTS), prior to acceptance. This applies to primary research papers only. ORCID helps the scientific community achieve unambiguous attribution of all scholarly contributions. You can create and link your ORCID from the home page of the MTS by clicking on 'Modify my Springer Nature account'. For more information please visit www.springernature.com/orcid.

We hope to receive your revised paper within 1-2 months. If you cannot send it within this time, please let us know.

Yours sincerely,

Reviewers Comments:

Reviewer #1 (Remarks to the Author):

Fiedler et al. have provided substantive updates to their manuscript, now titled "Dual-targeting 5-nitroimidazole ethers eradicate *H. pylori* by induction of stress and disability its adaptive response", that was previous reviewed for Nature. The revisions address key concerns raised by all reviewers and span all aspects of the paper, including additional in vitro and in vivo data. These updates increase the interpretability of the results and improve the manuscript. This work continues to be interesting and relevant for publication in Nature Microbiology. I will just re-raise a couple comments, though none rise to a level of concern at this point:

Figure 2, S6, S7, and S8 – The authors now provide a rationale for the focus on HpTpx and HpGroEL, which is helpful, and the totality of the data are consistent with their choice. However, I am still stuck on the possibility that there is more going on with some of these analogs suggesting additional targets could be explored to fully understand the activities of these molecules.

Figure 5C – The authors note in the figure legends that these are triple cocktail therapies, but it would be helpful to also capture this in the actual figure to avoid any confusion that these animals were not treated with Metro-P3, MF-01, or Standard Metro alone. Also, in the legends for this panel, the authors now note that the zero values are genuine (in response to a good catch by Reviewer #2). I am a little confused though: does this mean that the entirety of the stomach was plated and zero CFU were formed? If the entire organ was not plated, there should still be a limit of detection per gram based on the plated dilutions, correct?

Line 366 – Can the authors please add references for the work that previously synthesized these compounds (even if it was for other applications)?

Reviewer #2 (Remarks to the Author):

This revision partially addresses critiques, but major issues remain. According to the paper, the most significant findings are that the metronidazole analogs are 50 fold more effective than metronidazole in clearing an *H. pylori* infection in a mouse model; and that the analogs have a dual mode of action, in contrast to metronidazole. These assertions do not seem to be supported by the evidence presented.

Mechanism of action.

Responding to a previous critique, the authors note in the rebuttal: "It is important for us to underline that the main impact from this study results from deciphering the diverging MoAs of metronidazole and 5-nitroimidazole ethers, as this might have not been clear in our previous version: While both molecules induce oxidative stress and inhibit GroEL, the ether derivatives address Tpx as an additional target to block the stress response which is deleterious after its stress induction. To emphasize this unprecedented dual mode of action more clearly, we changed the title of the manuscript accordingly and carefully revised the main text."

- This is a puzzling statement. As the authors report: "Interestingly, when we performed the same concentration-dependent protein expression experiments with HpTpx in *E. coli*, Metro-P3 resulted in a 3.6-fold higher HpTpx modification compared to Metro (Figure 3F)." This result directly contradicts the assertion that the analogs have an additional target. Rather, the analogs have a modest 3.6 fold improvement in engaging a second target, Tpx, over metronidazole.

Animal efficacy.

Fig. 3 and Supplemental Fig. S 24 describing the efficacy data are poorly designed, presented and are challenging to understand. The X axis labels ostensibly depict the compounds that decrease the pathogen burden shown in bar graphs, such as "Metro-P3 1X". One assumes that this means that the decrease in pathogen shown in Metro-P3 1X bar is caused by Metro-P3 administration at some (not defined) 1X dose. This is not the case – Metro-P3 is actually part of a triple therapy regimen. The dose of Metro-P3, identity and dosage of those other compounds is not given in the figure legend. Since Metro-P3 and metronidazole are only administered as constituents of a triple therapy, drawing conclusions about their relative efficacy is uncertain. In a control, data with clarithromycin alone are presented. Metro-P3 and metronidazole should be similarly compared in the same experiment at the same doses as single compounds.

The authors specifically claim: "In the case of treatment with MF-01, already 0.30 mg/kg/day were sufficient for full *H. pylori* eradication, a dose at which Metro exhibited insufficient efficacy (Figure S24). This 50-fold reduction compared to Metro represents a major improvement in the treatment of *H. pylori* infections in vivo, significantly lowering drug exposure (Figure 5C)." However, they are not comparing MF-01 to a minimally efficacious dose of metronidazole, which remains to be determined. Rather, they compare MF-01 to a therapeutically recommended dose of metronidazole. Without knowing the minimally efficacious dose of metronidazole, it is impossible to draw any quantitative conclusions regarding the relative efficacy of these compounds. Note that experiment Fig. 5C is performed with Metro-P3, while S24 shows data for MF-01. This adds to the confusion.

Resistance.

In response to the critique, the authors perform an experiment with resistance development using sub-inhibitory levels of compounds and serial passage, and report that this resulted in the formation of round bodies. This is not informative. The standard test is frequency of resistance determination by plating pathogen on plates with various concentrations of drug starting at 4X MIC, and reporting the ratio of colonies grown/plated.

Recommendation.

The interesting observation of this study is the discovery of an additional target for the metronidazole series of compounds, Tpx. Focusing on the more compelling mechanistic studies and dropping inadequate microbiology and animal efficacy will improve the paper.

Reviewer #3 (Remarks to the Author):

In this extensive revision, the authors have responded to my initial review with substantial new data, including addition PK studies aimed at measuring exposure levels in relevant tissues, the synthesis and testing of additional control compounds, and repeat of proteomics studies with additional competitors. All of these studies add rigor to the results, which are surprising and significant, and should be of interest to the readers of *Nature Microbiology*. The response to the other reviewers comments and suggestions is similarly comprehensive. As such, I recommend publication of the revised manuscript and congratulate the authors on the work and their interesting findings.

Version 1:

Reviewer comments:

Reviewer #1

(Remarks to the Author)

In their most recent revision of "Dual-targeting 5-nitroimidazole ethers eradicate *H. pylori* by induction of stress and disabling its adaptive response", the authors have satisfactorily addressed all my current and previous comments and suggestions. As previously stated, this work will be of high general interest and is appropriate for publication in *Nature Microbiology*.

Reviewer #2

(Remarks to the Author)

This is a revised version of the manuscript that describes analogs of metronidazole with improved potency. The two main points the authors make is that, unlike metronidazole, the analogs engage an additional target; and that a particular analog is dramatically more efficacious than metronidazole. The evidence presented however does not support either of these conclusions.

As the authors note in their rebuttal: "We understand the reviewer's point that a claim of 50-fold reduction cannot be made if the minimal dosing of metronidazole is unknown.", and then state that finding the minimal efficacious dose of metronidazole in mice would have been unethical. Surprisingly though, they continue to present this unsubstantiated claim, with a heading describing the animal studies:

MF-01 shows 50-fold enhanced in vivo efficacy against *H. pylori* compared to Metro

Regarding the mechanism of action, the authors address the previous critique, included here in brackets ():

(Responding to a previous critique, the authors note in the rebuttal: "It is important for us to underline that the main impact from this study results from deciphering the diverging MoAs of metronidazole and 5-nitroimidazole ethers, as this might have not been clear in our previous version: While both molecules induce oxidative stress and inhibit GroEL, the ether derivatives address Tpx as an additional target to block the stress response which is deleterious after its stress induction. To emphasize this unprecedented dual mode of action more clearly, we changed the title of the manuscript accordingly and carefully revised the main text."

- This is a puzzling statement. As the authors report: "Interestingly, when we performed the same concentration-dependent protein expression experiments with HpTpx in *E. coli*, Metro-P3 resulted in a 3.6-fold higher HpTpx modification compared to Metro (Figure 3F)." This result directly contradicts the assertion that the analogs have an additional target. Rather, the analogs have a modest 3.6 fold improvement in engaging a second target, Tpx, over metronidazole.)

According to the current rebuttal:

"We thank the reviewer for pointing this out. We would like to clarify that we have two sets of experiments. One is the target identification in situ. Here, live cells were incubated with Metro-P1 and Metro-P3 (both ether-analogs exhibiting enhanced potency) or Metro-P2 (the metronidazole analog with low potency). Results from fluorescent SDS gels and MS both demonstrate that only Metro-P1 and Metro-P3 label and enrich Tpx while Metro-P2 does not (Figure 2). This is unequivocal proof of the described target selectivity."

- This is puzzling, since there is no mention of metronidazole, just a comparison between different analogs. The authors go on to state that conditions in *E. coli* where metronidazole labels Tpx are different from that of *H. pylori*. A simple interpretation is that metronidazole modifies Tpx to a lesser extent as compared to its analogs.

Decision Letter:

12th December 2025

Dear Stephan,

Thank you for your patience while your manuscript "Dual-targeting 5-nitroimidazole ethers eradicate *H. pylori* by induction of stress and disabling its adaptive response" was under peer-review at Nature Microbiology. It has now been seen by 2 referees, whose comments you will find at the end of this email. You will see from their comments below that while they find your work of interest, some important points are raised. We are very interested in the possibility of publishing your study in Nature Microbiology, but would like to consider your response to these concerns in the form of a revised manuscript before we make a final decision on publication.

In particular, you will see that while referee #1 is now satisfied with the study, referee #2 continues to have some important concerns. Referee #2 says you can't claim the derivatives are more efficacious than metronidazole because we don't know what the minimal dosing of Met is. Editorially, we will need you to remove this claim. Referee #2 also says you can't say that the derivatives have an additional target. The referee thinks that Met modifies Tpx to a lesser extent compared to its derivatives. This point will also need to be toned down. Editorially, we think that there are still other aspects that are interesting for our readership. The targets of metronidazole are currently unclear and this is a widely used antibiotic, so this information is of interest and will be important, plus you have derivatives that potentially have improved activity. However, as mentioned above, you will need to tone down aspects around improved activity compared to metronidazole. Framing could instead be identification of bacterial targets for metronidazole and its derivatives. The rest of the referees' reports are clear and the remaining issues should be straightforward to address.

If you have not done so already please begin to revise your manuscript so that it conforms to our Article format instructions at <http://www.nature.com/nmicrobiol/info/final-submission/>

The usual length limit for a Nature Microbiology Article is six display items (figures or tables) and 4,000 words. We have some flexibility, and can allow a revised manuscript at 4,500 words, but please consider this a firm upper limit. There is a trade-off of ~250 words per display item, so if you need more space, you could move a Figure or Table to Supplementary Information.

Some reduction could be achieved by focusing any introductory material and moving it to the start of your opening 'bold' paragraph, whose function is to outline the background to your work, describe in a sentence your new observations, and explain your main conclusions. The discussion should also be limited. Methods should be described in a separate section following the discussion, we do not place a word limit on Methods.

Nature Microbiology titles should give a sense of the main new findings of a manuscript, and should not contain punctuation. Please keep in mind that we strongly discourage active verbs in titles, and that they should ideally fit within 90 characters each (including spaces).

Please include a data availability statement as a separate section after Methods but before references, under the heading "Data Availability". This section should inform readers about the availability of the data used to support the conclusions of your study. This information includes accession codes to public repositories (data banks for protein, DNA or RNA sequences, microarray, proteomics data etc...), references to source data published alongside the paper, unique identifiers such as URLs to data repository entries, or data set DOIs, and any other statement about data availability. At a minimum, you should include the following statement: "The data that support the findings of this study are available from the corresponding author upon request", mentioning any restrictions on availability. If DOIs are provided, we also strongly encourage including these in the Reference list (authors, title, publisher (repository name), identifier, year). For more guidance on how to write this section please see: <http://www.nature.com/authors/policies/data/data-availability-statements-data-citations.pdf>

To improve the accessibility of your paper to readers from other research areas, please pay particular attention to the wording of the paper's opening bold paragraph, which serves both as an introduction and as a brief, non-technical summary in about 150 words. If, however, you require one or two extra sentences to explain your work clearly, please include them even if the paragraph is over-length as a result. The opening paragraph should not contain references. Because scientists from other sub-disciplines will be interested in your results and their implications, it is important to explain essential but specialised terms concisely. We suggest you show your summary paragraph to colleagues in other fields to uncover any problematic concepts.

If your paper is accepted for publication, we will edit your display items electronically so they conform to our house style and will reproduce clearly in print. If necessary, we will re-size figures to fit single or double column width. If your figures contain several parts, the parts should form a neat rectangle when assembled. Choosing the right electronic format at this stage will speed up the processing of your paper and give the best possible results in print. We would like the figures to be supplied as vector files - EPS, PDF, AI or postscript (PS) file formats (not raster or bitmap files), preferably generated with vector-graphics software (Adobe Illustrator for example). Please try to ensure that all figures are non-flattened and fully editable. All images should be at least 300 dpi resolution (when figures are scaled to approximately the size that they are to be printed at) and in RGB colour format. Please do not submit Jpeg or flattened TIFF files. Please see also 'Guidelines for Electronic Submission of Figures' at the end of this letter for further detail.

Figure legends must provide a brief description of the figure and the symbols used, within 350 words, including definitions of any error bars employed in the figures.

When submitting the revised version of your manuscript, please pay close attention to our [href="https://www.nature.com/nature-research/editorial-policies/image-integrity">Digital Image Integrity Guidelines](https://www.nature.com/nature-research/editorial-policies/image-integrity) and to the following points below:

EXTENDED DATA FIGURES

Please include a statement before the acknowledgements naming the author to whom correspondence and requests for materials should be addressed.

Finally, we require authors to include a statement of their individual contributions to the paper -- such as experimental work, project planning, data analysis, etc. -- immediately after the acknowledgements. The statement should be short, and refer to authors by their initials. For details please see the Authorship section of our joint Editorial policies at http://www.nature.com/authors/editorial_policies/authorship.html

* include a point-by-point response to any editorial suggestions and to our referees. Please include your response to the editorial suggestions in your cover letter, and please upload your response to the referees as a separate document.

* ensure it complies with our format requirements for Letters as set out in our guide to authors at www.nature.com/nmicrobiol/info/gta/

* state in a cover note the length of the text, methods and legends; the number of references; number and estimated final size of figures and tables

* resubmit electronically if possible using the link below to access your home page:

Link Redacted

*This url links to your confidential homepage and associated information about manuscripts you may have submitted or be reviewing for us. If you wish to forward this e-mail to co-authors, please delete this link to your homepage first.

Please ensure that all correspondence is marked with your Nature Microbiology reference number in the subject line.

Nature Microbiology is committed to improving transparency in authorship. As part of our efforts in this direction, we are now requesting that all authors identified as 'corresponding author' on published papers create and link their Open Researcher and Contributor Identifier (ORCID) with their account on the Manuscript Tracking System (MTS), prior to acceptance. This applies to primary research papers only. ORCID helps the scientific community achieve unambiguous attribution of all scholarly contributions. You can create and link your ORCID from the home page of the MTS by clicking on 'Modify my Springer Nature account'. For more information please visit [please visit www.springernature.com/orcid](http://www.springernature.com/orcid).

We hope to receive your revised paper within three weeks. If you cannot send it within this time, please let us know.

Yours sincerely,

Reviewers Comments:

Reviewer #1 (Remarks to the Author):

In their most recent revision of "Dual-targeting 5-nitroimidazole ethers eradicate *H. pylori* by induction of stress and disabling its adaptive response", the authors have satisfactorily addressed all my current and previous comments and suggestions. As previously stated, this work will be of high general interest and is appropriate for publication in Nature Microbiology.

Reviewer #2 (Remarks to the Author):

This is a revised version of the manuscript that describes analogs of metronidazole with improved potency. The two main points the authors make is that, unlike metronidazole, the analogs engage an additional target; and that a particular analog is dramatically more efficacious than metronidazole. The evidence presented however does not support either of these conclusions.

As the authors note in their rebuttal: "We understand the reviewer's point that a claim of 50-fold reduction cannot be made if the minimal dosing of metronidazole is unknown.", and then state that finding the minimal efficacious dose of metronidazole in mice would have been unethical. Surprisingly though, they continue to present this unsubstantiated claim, with a heading describing the animal studies:

MF-01 shows 50-fold enhanced in vivo efficacy against *H. pylori* compared to Metro

Regarding the mechanism of action, the authors address the previous critique, included here in brackets ():

(Responding to a previous critique, the authors note in the rebuttal: "It is important for us to underline that the main impact from this study results from deciphering the diverging MoAs of metronidazole and 5-nitroimidazole ethers, as this might have not been clear in our previous version: While both molecules induce oxidative stress and inhibit GroEL, the ether derivatives address Tpx as an additional target to block the stress response which is deleterious after its stress induction. To emphasize this unprecedented dual mode of action more clearly, we changed the title of the manuscript accordingly and carefully revised the main text."

- This is a puzzling statement. As the authors report: "Interestingly, when we performed the same concentration-dependent protein expression experiments with HpTpx in E. coli, Metro-P3 resulted in a 3.6-fold higher HpTpx modification compared to Metro (Figure 3F)." This result directly contradicts the assertion that the analogs have an additional target. Rather, the analogs have a modest 3.6 fold improvement in engaging a second target, Tpx, over metronidazole.)

According to the current rebuttal:

"We thank the reviewer for pointing this out. We would like to clarify that we have two sets of experiments. One is the target identification in situ. Here, live cells were incubated with Metro-P1 and Metro-P3 (both ether-analogs exhibiting enhanced potency) or Metro-P2 (the metronidazole analog with low potency). Results from fluorescent SDS gels and MS both demonstrate that only Metro-P1 and Metro-P3 label and enrich Tpx while Metro-P2 does not (Figure 2). This is unequivocal proof of the described target selectivity."

- This is puzzling, since there is no mention of metronidazole, just a comparison between different analogs. The authors go on to state that conditions in E. coli where metronidazole labels Tpx are different from that of H. pylori. A simple interpretation is that metronidazole modifies Tpx to a lesser extent as compared to its analogs.

Version 2:

Decision Letter:

21st December 2025

Dear Stephan,

Thank you for submitting your revised Article entitled "Metronidazole and ether derivatives eradicate H. pylori by induction of stress and disabling its adaptive response" for consideration. Editorially, we feel the paper has improved during revision and we will be happy to accept it in-principle. Before I can do so however, I would like to ask you to convert the following Supplementary Figures into Extended Data Figures (the figures with blots). This is so that our system can run its integrity screening for relevant figures. You will be able to choose different figures as Extended Data Figures (the maximum is 10) later on, once the checks are done.

Please convert Supplementary Figures 6, 7, 9, 10, 11, 12, 15, 16, 17, 21 into Extended Data Figures and resubmit using the link below.

When submitting the revised version of your manuscript, please pay close attention to our [href="https://www.nature.com/nature-research/editorial-policies/image-integrity">Digital Image Integrity Guidelines. and to the following points below:](https://www.nature.com/nature-research/editorial-policies/image-integrity)

Link Redacted

I'm sending you this message already now in case you wish to address this before our office re-opens. The Nature Microbiology editorial office will be closed from Dec 22 - Jan 2. I'll be back in the office on Jan 5 and will then prioritize handling your study. Once we receive the submission files (including the 10 Extended Data Figures mentioned above), I will send the official Accept-in-principle decision letter. After that we will prepare a checklist for you to address some further editorial and formatting points, but this will be in early/mid January.

Best wishes and Happy Holidays,

Version 3:

Decision Letter:

Our ref: NMICROBIOL-25082995C

12th January 2026

Dear Stephan,

Thank you for submitting your revised manuscript "Metronidazole and ether derivatives eradicate *H. pylori* by induction of stress and disabling its adaptive response" (NMICROBIOL-25082995C). Editorially, we find that the paper has improved in revision, and therefore we'll be happy in principle to publish it in Nature Microbiology, pending minor revisions to comply with our editorial and formatting guidelines.

Thank you again for your interest in Nature Microbiology. Please do not hesitate to contact me if you have any questions.

Sincerely,

Version 4:

Decision Letter:

11th February 2026

Dear Stephan,

I am pleased to accept your Article "Metronidazole and ether derivatives target *Helicobacter pylori* via simultaneous stress induction and inhibitor" for publication in Nature Microbiology. Thank you for having chosen to submit your work to us and many congratulations.

Authors may need to take specific actions to achieve compliance with funder and institutional open access mandates. If your research is supported by a funder that requires immediate open access (e.g. according to [a href="https://www.springernature.com/gp/open-science/plan-s-compliance"](https://www.springernature.com/gp/open-science/plan-s-compliance) Plan S principles or the a href="https://www.springernature.com/gp/open-science/plan-s-compliance"

<https://www.springernature.com/gp/open-science/us-federal-agency-compliance>) then you should select the gold OA route, and we will direct you to the compliant route where possible. Because authors warrant under our subscription licensing terms that they haven't committed to licensing any version of their article under a licence inconsistent with the terms of our agreement – including the applicable embargo period – publication under the subscription model isn't suitable for authors whose funders require no embargo.

Congratulations once again and I look forward to seeing the article published.

With kind regards,

P.S. Click on the following link if you would like to recommend Nature Microbiology to your librarian
<http://www.nature.com/subscriptions/recommend.html#forms>

** Visit the Springer Nature Editorial and Publishing website at http://editorial-jobs.springernature.com?utm_source=ejP_NMicro_email&utm_medium=ejP_NMicro_email&utm_campaign=ejP_NMicro for more information about our career opportunities. If you have any questions please click [here](mailto:editorial.publishing.jobs@springernature.com).

Garching b. München, 7. November 2025

Revised version of NMICROBIOL-25082995-T

Point-to-point response:

Reviewer 1:

Figure 2, S6, S7, and S8 – The authors now provide a rationale for the focus on HpTpx and HpGroEL, which is helpful, and the totality of the data are consistent with their choice. However, I am still stuck on the possibility that there is more going on with some of these analogs suggesting additional targets could be explored to fully understand the activities of these molecules.

The reviewer is right with the statement that, as with any other drug, additional targets cannot be excluded. However, we here used state of the art target identification techniques including activity-based protein profiling combined with binding site studies (iso-DTB) and experimentally validate our findings. Although unlikely, we cannot exclude binding to DNA or metabolites as these cannot be assessed by proteomics. To address this concern, we added a sentence in the discussion explaining this point.

Figure 5C – The authors note in the figure legends that these are triple cocktail therapies, but it would be helpful to also capture this in the actual figure to avoid any confusion that these animals were not treated with Metro-P3, MF-01, or Standard Metro alone. Also, in the legends for this panel, the authors now note that the zero values are genuine (in response to a good catch by Reviewer #2). I am a little confused though: does this mean that the entirety of the stomach was plated and zero CFU were formed? If the entire organ was not plated, there should still be a limit of detection per gram based on the plated dilutions, correct?

We thank the reviewer for this comment. We agree that the bar graphs in the figure are poorly annotated and corrected this accordingly. Regarding the zero values, indeed, not the entire stomach was plated, and therefore a (low) limit of detection applies. As a horizontal line indicating the LOD would not be visible in the figure, we included this information in the figure description and added the corresponding calculation to the Methods section for clarity.

Line 366 – Can the authors please add references for the work that previously synthesized these compounds (even if it was for other applications)?

The reference was added.

Reviewer 2:

Mechanism of action.

Responding to a previous critique, the authors note in the rebuttal: “It is important for us to underline that the main impact from this study results from deciphering the diverging MoAs of metronidazole and 5-nitroimidazole ethers, as this might have not been clear in our previous version: While both molecules induce oxidative stress and inhibit GroEL, the ether derivatives address Tpx as an additional target to block the stress response which is deleterious after its stress induction. To emphasize this unprecedented dual mode of action more clearly, we changed the title of the manuscript accordingly and carefully revised the main text.”

- This is a puzzling statement. As the authors report: “Interestingly, when we performed the same concentration-dependent protein expression experiments with HpTpx in *E. coli*, Metro-P3 resulted in a 3.6-fold higher HpTpx modification compared to Metro (Figure 3F).” This result directly contradicts the assertion that the analogs have an additional target. Rather, the analogs have a modest 3.6 fold improvement in engaging a second target, Tpx, over metronidazole.

We thank the reviewer for pointing this out. We would like to clarify that we have two sets of experiments. One is the target identification *in situ*. Here, live cells were incubated with Metro-P1 and Metro-P3 (both ether-analogs exhibiting enhanced potency) or Metro-P2 (the metronidazole analog with low potency). Results from fluorescent SDS gels and MS both demonstrate that only Metro-P1 and Metro-P3 label and enrich Tpx while Metro-P2 does not (Figure 2). This is unequivocal proof of the described target selectivity.

The second set is done for validation with overexpressed Tpx in *E. coli* cells which is physiologically different from the situation in living *H. pylori*. Higher compound concentrations were needed to achieve binding to the protein and thus resulted in modification of Tpx also by metronidazole. In this experimental setup, an 8-fold higher concentration of metronidazole was needed to achieve the same extent of peroxidase inhibition, demonstrating a notable difference also under these conditions. It is not surprising that compounds modify enzymes at higher concentrations which are not their primary targets, a strategy that is used often e.g. in crystallography. In fact, we deliberately modified Tpx with metronidazole to demonstrate its impaired binding in comparison to Metro-P3 via a co-crystal structure.

To clarify, we wrote “However, as the overexpression of target proteins leads to higher levels compared to the endogenous system, elevated compound concentrations were required for binding. Despite this limitation, the methodology turned out to be robust, demonstrating that

HpGroEL was bound and inhibited by **Metro** and the ether analog **Metro-P3** in their activated forms to a similar extent.” in the discussion.

Animal efficacy.

Fig. 3 and Supplemental Fig. S 24 describing the efficacy data are poorly designed, presented and are challenging to understand. The X axis labels ostensibly depict the compounds that decrease the pathogen burden shown in bar graphs, such as “Metro-P3 1X”. One assumes that this means that the decrease in pathogen shown in Metro-P3 1X bar is caused by Metro-P3 administration at some (not defined) 1X dose. This is not the case – Metro-P3 is actually part of a triple therapy regiment. The dose of Metro-P3, identity and dosage of those other compounds is not given in the figure legend. Since Metro-P3 and metronidazole are only administered as constituents of a triple therapy, drawing conclusions about their relative efficacy is uncertain. In a control, data with clarithromycin alone are presented. Metro-P3 and metronidazole should be similarly compared in the same experiment at the same doses as single compounds.

We thank the reviewer for this important comment and would like to apologize for the confusion. The labeling of the bars in Figure 5 C and S24 is indeed insufficient and was replaced by appropriate legends including the triple therapy in addition to dosing information in the figure legend.

The reviewer claims that comparing the effect of metronidazole and the ether analog as part of a triple therapy is “uncertain” without giving any explanation why this should be the case. If all experimental conditions remain the same including the concentrations of clarithromycin and omeprazole, why would it not be possible to compare two compounds? This is routinely done in combination therapies for many diseases. This argumentation is not only difficult to follow but also beyond what our federal authorities for animal studies will accept. In response to the reviewer’s request, we have already inquired about this issue with our institution’s animal welfare officer, who straightly rejected this idea.

Metronidazole is used in triple or even quadruple therapies representing the state of the art and must thus be the set standard. The testing of clarithromycin and omeprazole as control treatment was part of this study as background control, which is allowed by our animal legislation in order to evaluate the effect size of a new treatment, a standard approach in drug development for co-therapies. We understand that this might have been confusing and removed the clarithromycin/omeprazole control data completely to avoid confusion.

The authors specifically claim: “In the case of treatment with MF-01, already 0.30 mg/kg/day were sufficient for full H. pylori eradication, a dose at which Metro exhibited insufficient efficacy (Figure S24). This 50-fold reduction compared to Metro represents a major improvement in the treatment of H. pylori infections in vivo, significantly lowering drug exposure (Figure 5C).” However, they are not comparing MF-01 to a minimally efficacious dose of metronidazole, which remains to be determined. Rather, they compare MF-01 to a therapeutically recommended dose of metronidazole. Without knowing the minimally efficacious dose of metronidazole, it is impossible to draw any quantitative conclusions regarding the relative efficacy of these compounds. Note that experiment Fig. 5C is performed with Metro-P3, while S24 shows data for MF-01. This adds to the confusion.

We understand the reviewer's point that a claim of 50-fold reduction cannot be made if the minimal dosing of metronidazole is unknown. Finding out this dose would require the titration of the drug in animals. Given the established treatment regimen of metronidazole for *H. pylori* we will again not get any permission to conduct such titration studies with numerous animals and agree that this would be unethical. We would like to remind that we performed, in response to comments of this and other reviewers during the first revision, additional studies with metronidazole at the same 0.3 mg/kg dose to compare it with the ether analog and did not see any effect. In line with the reviewer comment, we stated in the revision that at a 0.3 mg/kg dosing of MF-01 within a triple therapy leads to full eradication of *H. pylori* while the same dosing of metronidazole showed no effect.

Resistance.

In response to the critique, the authors perform an experiment with resistance development using sub-inhibitory levels of compounds and serial passage, and report that this resulted in the formation of round bodies. This is not informative. The standard test is frequency of resistance determination by plating pathogen on plates with various concentrations of drug starting at 4X MIC, and reporting the ratio of colonies grown/plated.

We searched the literature and it is quite stunning that there is no publically available FoR data for metronidazole in *H. pylori*. Nevertheless, in the response to this comment, we performed frequency of resistance (FoR) studies, with metronidazole, Metro-P3 and MF-01 at various concentrations. 8x MIC (MF-01) and 12x MIC (Metro-P3) showed a FoR of $< 7 \times 10^{-8}$ comparable to the effect of 4x MIC of metronidazol. This corresponds to concentrations of 3.04 μM and 6.24 μM for the ether analogs opposed to 50 μM for metronidazol. Still, as lower concentrations of the ether analogs caused resistance development it contrasts our results for the serial passaging assay where MF-01 did not show any MIC shifts and Metro-P3 was lower compared to metronidazole. While we can only speculate about the different outcomes of these two assays (e.g. differences in compound stability/solubility, especially when embedded in hot agar and different cultivation settings), we will be cautious in our analysis and write "Serial passaging showed no pronounced resistance development of 5-nitroimidazole ethers. Frequency of resistance (FoR) studies with **Metro-P3** at 12x MIC confirmed this notion (FoR $< 2.01 \times 10^{-11}$), however, notable resistance occurred at 8x MIC (1.52 μM) (**Table S1, Figure S2**)."

We thank the reviewers for their time and thoughtful comments which helped to further improve this manuscript.

Best wishes,

Stephan Sieber on behalf of all authors

Garching b. München, 16. Dezember 2025

Revised version of NMICROBIOL-25082995-T

Thank you for your constructive suggestions to resolve the remaining concerns. I agree that the story is equally exciting by highlighting the novel targets of metronidazole which provides unprecedented insights into a so far unknown mechanism of action.

To address your suggestions, the following changes were incorporated:

1. New title including metronidazole: "Metronidazole and ether derivatives eradicate *H. pylori* by induction of stress and disabling its adaptive response"
2. The abstract has been re-written to include target identification for metronidazole:
Metronidazole represents a stress inducing front line-drug in the treatment of *Helicobacter pylori* infections. To unravel its poorly defined cellular targets, we performed activity-based protein profiling (ABPP) with tailored metronidazole probes. ABPP revealed chaperonin HpGroEL and thiol peroxidase HpTpx as prominent targets, the latter being essential for *H. pylori* survival under oxidative stress. Surprisingly, alkynylated ether probes exhibited a 60-fold enhanced antibacterial potency compared to the parent drug. Biological assays and co-crystallization studies confirmed target engagement, with an enhanced binding of ether derivatives to HpTpx. Thus, induction of stress and simultaneous inhibition of the stress response represents a so far overlooked mechanism of this compound class. Refined ether analogs exhibited favorable pharmacological profiles without cytotoxicity. The activity boost translated to an *in vivo H. pylori* mouse model demonstrating full eradication of bacteria at low dosing of 0.3 mg/kg/day.
3. The results section is largely unchanged. As requested, the headline stating the 50-fold enhanced *in vivo* potency was changed and now reads: "MF-01 shows excellent *in vivo* efficacy against *H. pylori* at low dosing". Moreover, we adjusted Figure 5D and took out the statement of 50-fold potency enhancement.
4. The conclusion section was reworked to highlight a dual mode of action discovered for metronidazole.

All these changes are highlighted in the manuscript. We are confident that this version addresses all remaining concerns.

We thank you for your time and thoughtful comments which helped to further improve this manuscript.

Best wishes,

Stephan Sieber on behalf of all authors